RESEARCH ARTICLE  

# Intrinsic excitation-inhibition imbalance affects medial prefrontal cortex differently in autistic men versus women

Stavros Trakoshis[1,2†], Pablo Martínez-Cañada[3,4†], Federico Rocchi[5],
Carola Canella[5], Wonsang You[6], Bhismadev Chakrabarti[7,8], Amber NV Ruigrok[7],
Edward T Bullmore[9,10], John Suckling[10], Marija Markicevic[11,12], Valerio Zerbi[11,12],
MRC AIMS Consortium, Simon Baron-Cohen[7,9], Alessandro Gozzi[5‡],
Meng-Chuan Lai[7,13,14,15,16‡], Stefano Panzeri[3‡], Michael V Lombardo[1,7‡*]

[1]Laboratory for Autism and Neurodevelopmental Disorders, Center for Neuroscience and Cognitive Systems @UniTn, Istituto Italiano di Tecnologia, Rovereto, Italy; [2]Department of Psychology, University of Cyprus, Nicosia, Cyprus; [3]Neural Computation Laboratory, Center for Neuroscience and Cognitive Systems @UniTn, Istituto Italiano di Tecnologia, Rovereto, Italy; [4]Optical Approaches to Brain Function Laboratory, Department of Neuroscience and Brain Technologies, Istituto Italiano di Tecnologia, Genova, Italy; [5]Functional Neuroimaging Laboratory, Center for Neuroscience and Cognitive Systems @UniTn, Istituto Italiano di Tecnologia, Rovereto, Italy; [6]Artificial Intelligence and Image Processing Laboratory, Department of Information and Communications Engineering, Sun Moon University, Asan, Republic of Korea; [7]Autism Research Centre, Department of Psychiatry, University of Cambridge, Cambridge, United Kingdom; [8]Centre for Integrative Neuroscience and Neurodynamics, School of Psychology and Clinical Language Sciences, University of Reading, Reading, United Kingdom; [9]Cambridgeshire and Peterborough National Health Service Foundation Trust, Cambridge, United Kingdom; [10]Brain Mapping Unit, Department of Psychiatry, University of Cambridge, Cambridge, United Kingdom; [11]Neural Control of Movement Lab, D-HEST, ETH Zurich, Zurich, Switzerland; [12]Neuroscience Center Zurich, University and ETH Zurich, Zurich, Switzerland; [13]The Margaret and Wallace McCain Centre for Child, Youth & Family Mental Health, Azrieli Adult Neurodevelopmental Centre, and Campbell Family Mental Health Research Institute, Centre for Addiction and Mental Health, Toronto, Canada; [14]Department of Psychiatry and Autism Research Unit, The Hospital for Sick Children, Toronto, Canada; [15]Department of Psychiatry, Faculty of Medicine, University of Toronto, Toronto, Canada; [16]Department of Psychiatry, National Taiwan University Hospital and College of Medicine, Taipei, Taiwan

**\*For correspondence:**
mvlombardo@gmail.com

[†]These authors contributed equally to this work
[‡]These authors also contributed equally to this work

**Abstract** Excitation-inhibition (E:I) imbalance is theorized as an important pathophysiological mechanism in autism. Autism affects males more frequently than females and sex-related mechanisms (e.g., X-linked genes, androgen hormones) can influence E:I balance. This suggests that E:I imbalance may affect autism differently in males versus females. With a combination of in-silico modeling and in-vivo chemogenetic manipulations in mice, we first show that a time-series metric estimated from fMRI BOLD signal, the Hurst exponent (H), can be an index for underlying change in the synaptic E:I ratio. In autism we find that H is reduced, indicating increased excitation, in the medial prefrontal cortex (MPFC) of autistic males but not females. Increasingly intact MPFC

H is also associated with heightened ability to behaviorally camouflage social-communicative difficulties, but only in autistic females. This work suggests that H in BOLD can index synaptic E:I ratio and that E:I imbalance affects autistic males and females differently.

## Introduction

Excitation-inhibition (E:I) balance in the brain has been hypothesized to be atypical in many neuro-psychiatric conditions (*Rubenstein and Merzenich, 2003*; *Sohal and Rubenstein, 2019*), including autism. Rubenstein and Merzenich originally suggested that some types of autism may be explained by an E:I imbalance that may lead to hyper-excitability in cortical circuitry and potentially enhanced levels of neuronal noise (*Rubenstein and Merzenich, 2003*). However, coming to a better understanding of how E:I balance is affected across a heterogeneous mixture of autistic individuals has proven to be challenging because of the limited availability of robust E:I biomarkers that are non-invasive and applicable in humans and which can be measured on a large scale. A majority of the literature about E:I balance in autism extends from investigations of prominent single gene mutations associated with autism and the animal model research around these genes (*Sohal and Rubenstein, 2019*; *Nelson and Valakh, 2015*). This leaves a significant gap in evaluating the E:I theory on a larger majority of the autistic population. While no one theory can fully explain all individuals with an autism diagnosis (*Happé et al., 2006*; *Lombardo et al., 2019a*), the E:I imbalance theory may have utility for understanding subtypes of autistic individuals (*Lombardo et al., 2015*; *Lombardo et al., 2018a*; *Lombardo et al., 2019b*).

Sex/gender may be an important stratifier of relevance for highlighting E:I imbalance subtypes (*Lai et al., 2015*; *Lai et al., 2017a*). Many highly penetrant autism-associated genes are located on the sex chromosomes (e.g., *FMR1*, *MECP2*, *NLGN3*, *GABRA3*, *ARX*, *SYN1*) and are known to lead to pathophysiology implicating E:I dysregulation (*Rubenstein and Merzenich, 2003*; *Lee et al., 2017*; *Bozzi et al., 2018*). Other genes playing important roles in the balance between excitation and inhibition in the brain (e.g., *MEF2C*, *GRIK2*, *GRIA1*, *SCN3A*, *SCN9A*, *NPTX2*) are highly sensitive to androgens in human neuronal stem cells and are highly expressed in 'social brain' circuitry such as the default mode network, and in particular, the medial prefrontal cortex (MPFC) (*Lombardo et al., 2018b*). Optogenetic stimulation to enhance excitation in mouse MPFC also results in changes in social behavior (*Yizhar et al., 2011*; *Selimbeyoglu et al., 2017*). These results hint that sex-relevant biological mechanisms affect E:I balance and that key social brain regions such as MPFC may be of particular importance for explaining how E:I imbalance affects social behavior. Sex/gender heterogeneity also leads to differing clinical presentations and compensatory mechanisms in autism that may depend on E:I balance in MPFC. It is known that many cognitively able adult autistic women engage in camouflaging behaviors that tend to compensate or mask their social-communicative difficulties moreso than autistic men (*Lai et al., 2017b*; *Hull et al., 2020*; *Schuck et al., 2019*). Prior work has shown that whereas autistic males show reduced ventral MPFC (vMPFC) self-representation response, autistic females show intact vMPFC self-representation. Furthermore, the degree to which vMPFC shows intact self-representation response in autistic females is associated with enhanced ability to camouflage (*Lai et al., 2019*). If E:I imbalance asymmetrically affects vMPFC function in males versus females, this could help explain differential camouflaging in adult autistic females.

To better understand sex-specific E:I imbalance in autism we need better neuroimaging biomarkers that index underlying synaptic E:I mechanisms and which can be deployed on a large-scale for in-vivo investigation in deeply phenotyped cohorts. Here we pursue the idea that spectral properties of neural time-series data (e.g., local field potentials (LFP) or blood oxygen level dependent (BOLD) signal) could be used to isolate such biomarkers. It has been long known that LFP and resting state fMRI (rsfMRI) data exhibits rich spectral properties, with power decreasing as function of frequency (*Bullmore et al., 2001*; *Maxim et al., 2005*; *He, 2011*; *Bédard et al., 2006*; *He, 2014*). Models of neural networks have reported that the E:I ratio has profound effects on the spectral shape of electrophysiological activity (*Brunel and Wang, 2003*; *Mazzoni et al., 2008*; *Lombardi et al., 2017*). Recent work with simplified models has proposed that the exponent of the 1/f spectral power law, an index closely related to the Hurst exponent (H), reflects the extent of E:I imbalance (*Gao et al., 2017*). This suggests that neurophysiologically heightened E:I ratio generates flatter 1/f slope in LFP data and this could drive H (as measured in BOLD) to be decreased. In past

**eLife digest** Autism is a condition that is usually diagnosed early in life that affects how a person communicates and socializes, and is often characterized by repetitive behaviors. One key theory of autism is that it reflects an imbalance in levels of excitation and inhibition in the brain. Excitatory signals are those that make other brain cells more likely to become active; inhibitory signals have the opposite effect. In non-autistic individuals, inhibitory activity outweighs excitatory activity. In people with autism, by contrast, an increase in excitatory activity is believed to produce an imbalance in excitation and inhibition.

Most of the evidence to support this excitation-inhibition imbalance theory has come from studies of rare mutations that cause autism. Many of these mutations occur on the sex chromosomes or are influenced by androgen hormones (hormones that usually play a role on typically male traits). However, most people with autism do not possess these particular mutations. It was thus unclear whether the theory could apply to everyone with autism or, for example, whether it may better apply to specific groups of individuals based on their sex or gender. This is especially important given that about four times as many men and boys compared to women and girls are diagnosed with autism.

Trakoshis, Martínez-Cañada et al. have now found a way to ask whether any imbalance in excitation and inhibition in the brain occurs differently in men and women. Using computer modeling, they identified a signal in brain scans that corresponds to an imbalance of excitation and inhibition. After showing that the technique works to identify real increases in excitation in the brain scans of mice, Trakoshis, Martínez-Cañada et al. looked for this signal, or biomarker, in brain scans of people with and without autism. All the people in the study identified with the gender that matched the sex they were assigned at birth. The results revealed differences between the men and women with autism. Men with autism showed an imbalance in excitation and inhibition in specific 'social brain' regions including the medial prefrontal cortex, but women with autism did not. Notably, many of these brain regions are strongly affected by androgen hormones.

Previous studies have found that women with autism are sometimes better at hiding or 'camouflaging' their difficulties when socializing or communicating than men with autism. Trakoshis, Martínez-Cañada et al. showed that the better a woman was at camouflaging her autism, the more her brain activity in this region resembled that of non-autistic women.

Excitation-inhibition imbalance may thus affect specific brain regions involved in socializing and communication more in men who have autism than in women with the condition. Balanced excitation and inhibition in these brain areas may enable some women with autism to camouflage their difficulties socializing or communicating. Being able to detect imbalances in activity using standard brain imaging could be useful for clinical trials. Future studies could use this biomarker to monitor responses to drug treatments that aim to adjust the balance between excitation and inhibition.

work we have shown that H is atypically decreased in rsfMRI data of adult autistic males, particularly for social brain areas like MPFC (*Lai et al., 2010*). H is statistically relevant for the concept of 'neural noise' since lower levels of H can be interpreted as closer to what would be expected of a completely noisy random signal (e.g., white noise produces an H = 0.5). Related to H and long-memory characteristics of the rsfMRI time-series, prior work has also shown case-control differences in the intrinsic neural timescale in autism (e.g., magnitude of temporal autocorrelation) (*Watanabe et al., 2019*). However, these prior studies examine primarily male-dominated samples and thus cannot shed insight into sex-related heterogeneity in autism.

In this work we aim to better understand how E:I imbalance may differentially affect autistic males and females. To achieve this aim, we first took a bottom-up approach by using in-silico (i.e. computational) models of local neuronal microcircuitry to make predictions about how H and 1/f slope in LFP and rsfMRI BOLD data may behave when there are underlying changes in E:I balance. Importantly, our approach takes a major step forward from prior work (*Gao et al., 2017*) by utilizing a model that includes interactions within and between excitatory and inhibitory neuronal populations. Next, our in-silico predictions are then tested in-vivo with a combination of rsfMRI and experimental

manipulations in mice that either increase neurophysiological excitation or that silence the local activity in the network. Chemogenetic (i.e. designer receptors exclusively activated by designer drugs; DREADD) or optogenetic manipulations are optimally suited to these purposes, owing to the possibility of enabling remote control of neuronal excitability with cell-type and regional specificity (*Yizhar et al., 2011*; *Ferenczi et al., 2016*). Manipulations of neuronal activity like these in animals are key for two reasons. First, they allow for experimental in-vivo confirmation of in-silico predictions. Second, such work is a key translational link across species (i.e. rodents to humans), given the common use of neuroimaging readouts from rsfMRI (*Balsters et al., 2020*). At the genomic level we then examine what cell types could possibly underlie sex-related heterogeneity in E:I imbalance. Finally, we then turn to the human rsfMRI data to show how E:I imbalance may differ amongst autistic males and females and how such mechanisms may explain individual differences in camouflaging behavior.

## Results

### Analysis of E:I balance in simulated LFPs from a recurrent network model

In a bottom-up fashion, we first worked to identify potential biomarkers of E:I imbalance from neural time-series data such as local field potentials (LFPs). Motivating our in-silico modeling of E:I effects on LFP and BOLD data, we note prior work by Gao and colleagues (*Gao et al., 2017*). This prior work simulated LFP time-series from non-interacting excitatory and inhibitory neuronal populations (*Figure 1—figure supplement 1A*) and showed that spectral properties such as the 1/f slope flatten with increasing E:I ratio (*Figure 1—figure supplement 1B*). Given the relationship between 1/f slope and H (*Stadnitski, 2012*), we show within this modeling approach that as E:I ratio increases, H decreases (*Figure 1—figure supplement 1C*). However, a limitation of this prior work is that it does not include interactions between excitatory and inhibitory populations nor does it allow for recurrent connections within such populations.

To address these limitations, we developed a more biologically plausible recurrent network model of interacting excitatory and inhibitory integrate-and-fire neuronal populations that receive external inputs (both a sensory driven thalamic input and a sensory unrelated intracortical input) (*Figure 1A*; see Materials and methods for more details). From this model, we computed the network's LFP as the sum of absolute values of all synaptic currents. The absolute value is taken because AMPA synapses are usually apical and GABA synapse are peri-somatic and thus their dipoles sum with the same sign along dendrites (*Mazzoni et al., 2008*; *Mazzoni et al., 2010*; *Deco et al., 2004*). We computed LFP summing presynaptic currents from both external inputs and recurrent interactions, as real LFPs capture both sources of synaptic activity (*Logothetis, 2008*). We have extensively validated this method of computing LFPs from integrate-and-fire networks in previous work on both real cortical data and simulations with networks of realistically-shaped 3D neurons and have shown that it works better than when using alternatives such as the sum of simulated membrane potentials, the signed sum of synaptic currents or a time integration of the spike rate (*Mazzoni et al., 2008*; *Mazzoni et al., 2015*). In this *in-silico* network, we manipulated the E:I ratio by independently varying the strengths of the inhibitory ($g_I$) and excitatory ($g_E$) synaptic conductances. We called $g$ the relative ratio between inhibitory and excitatory conductances ($g = g_I/g_E$). We report simulation results for two levels of strength of thalamic input ($v_0$ = 1.5 spikes/second and $v_0$ = 2 spikes/second), and we verified that our results hold qualitatively for a wider range of input levels (1.5 to 4 spikes/second).

*Figures 1B-C* show examples of LFP time-series and power spectral densities (PSDs) for two values of $g$, one within an excitation-dominated regime ($g$ = 5.6) and the other within an inhibition-dominated regime ($g$ = 14.8). The spectral profiles (*Figure 1C*) display two different regions of frequencies with different spectral properties: a region of steeper negative 1/f slopes at higher frequencies (> 30 Hz) and a region of shallower (small negative and sometimes positive) slopes at low frequencies (< 30 Hz). Thus, we calculated slopes for the low- and high-frequency regions with piecewise regressions of log power predicted by log frequency. Slopes from the low-frequency (*Figure 1D*) and high-frequency region (*Figure 1E*) increase when $g$ is reduced (i.e. E:I ratio augmented). This means that lower values of $g$ correspond to faster spectra with relatively more power



**Figure 1.** Predictions from a recurrent network model of how the low- and high-frequency slopes of the LFP power spectrum and H change with the variation in relative ratio of inhibitory and excitatory synaptic conductances. Panel **A** shows a sketch of the point-neuron network that includes recurrent connections between two types of populations: excitatory cells (E) and inhibitory cells (I). Each population receives two types of external inputs: intracortical activity and thalamic stimulation. Panels **B** and **C** show examples of normalized LFP times-series and their corresponding PSDs generated for two different ratios between inhibitory and excitatory conductances ($g = g_I/g_E$). The low- and high-frequency slopes of the piecewise regression lines that fit the log-log plot of the LFP PSDs are computed over two different frequency ranges (1-30 Hz for the low-frequency slope and 30-100 Hz for the high-frequency slope). The relationship between low-frequency slopes (panel **D**), high-frequency slopes (panel **E**) and H values (panel **F**) are plotted as a function of $g$ for two different firing rates of thalamic input (1.5 and 2 spikes/second). The reference value of $g$ (which has shown in previous studies to reproduce cortical data well) is represented by a dashed black line. In panel **G** and **H**, we show H values in 3 different groups of $g$ (high, medium and low $g$), with the same number of samples in each group.

The online version of this article includes the following figure supplement(s) for figure 1:

**Figure supplement 1.** In silico predictions from a non-recurrent model of how 1/f slope and H change with E:I ratio ($g$).

**Figure supplement 2.** 1/f slope estimation with the FOOOF algorithm.

at higher frequencies. Changes in slopes are more prominent in the excitation-dominated region where $g$ is smaller (that is, E:I ratio is shifted in favor of E) than the reference value ($g$ = 11.3), which has been shown to be a plausible reference value that reproduces cortical power spectra well (*Mazzoni et al., 2008*; *Mazzoni et al., 2010*; *Mazzoni et al., 2015*; *Mazzoni et al., 2011*; *Barbieri et al., 2014*; *Cavallari et al., 2014*). An increase in $g$ beyond this reference value (shifting the E:I balance towards stronger inhibition) had a weaker effect on slopes. Similar results were obtained when quantifying 1/f slope using the FOOOF algorithm (*Haller, 2018*; *Figure 1—figure supplement 2*), indicating that slopes are not biased by the particular piecewise linear fit procedure.

Next, we computed H from the same simulated LFPs. As expected, H decreases with decreasing $g$ (i.e. increasing E:I ratio), but only when $g$ is below the baseline reference value (*Figure 1F*). These results clearly indicate that, in a biologically plausible computational model of local cortical microcircuitry including recurrent connections between excitatory and inhibitory neuronal populations, changes in synaptic E:I ratio are reflected by and thus could be inferred from the overall LFP readout of 1/f slope or H.

## Simulated BOLD signal tracks with changes in E:I ratio and correlates selectively with LFP power bands

Given that E:I ratio in LFP data is related to 1/f slope and H, we next asked whether simulated fMRI BOLD signal from the recurrent model would also show similar relationships. To answer this question, we first had to simulate BOLD data from the LFP data generated from the recurrent model. Our approach to simulating BOLD (see Materials and methods and *Figure 2—figure supplement 1* for how BOLD was simulated from LFP), captures several key characteristics about the empirical relationship between LFP and BOLD. Studies with simultaneous LFP and BOLD measured in animals have shown that although BOLD signal correlates with both LFPs and spikes, it correlates more strongly with the LFP than with spikes (*Logothetis et al., 2001*; *Magri et al., 2012*; *Rauch et al., 2008*; *Viswanathan and Freeman, 2007*; *Lauritzen and Gold, 2003*). Further studies with simultaneous LFP and BOLD measured in non-human primates (*Logothetis et al., 2001*; *Magri et al., 2012*; *Schölvinck et al., 2010*) have considered the relationship between frequency-resolved LFPs and BOLD and indicate that LFP power shows time-lagged correlations with the time course of BOLD signal and that different frequency bands vary in how they correlate with measured BOLD signal. In particular, gamma band frequencies tend to show the strongest correlation between LFP power and BOLD signal. Frequency dependency of the EEG-BOLD relationship, with prominent predictive power of the gamma band, is also reported in humans (*Scheeringa et al., 2011*). Remarkably, these empirical observations are recapitulated with simulated LFP and BOLD data from the recurrent model. *Figure 2A* shows time-lagged correlations between time-dependent LFP power and BOLD. *Figure 2B* shows that all considered LFP frequency bands (e.g., alpha, beta, gamma) correlate with BOLD, but with the gamma band showing the strongest correlations. Thus, our method for simulating BOLD from recurrent model LFP data retains key empirical relationships observed between real LFP power and BOLD. Simulating BOLD with a simple hemodynamic response function (HRF) convolution of the LFP would have not respected the patterns of correlations between LFP power and BOLD observed in empirical data (i.e. the relative increase in correlation between the gamma band and BOLD with respect to other bands; *Figure 2—figure supplement 2*).

With simulated BOLD from the recurrent model, we next computed H on these data to understand if E:I ratio in the recurrent model is associated with changes in H in BOLD. Strikingly, H in BOLD shows the same dependency on $g$ as observed in LFP data (*Figure 2C-D*) - H in BOLD decreases as E:I ratio is shifted toward higher excitation by lowering the value of $g$ with respect to the reference value. Although H in LFP and BOLD showed similar associations with respect to changes in $g$, it is notable that the range of H in BOLD is shifted towards smaller values (*Figure 2C-D*) than H in LFP (*Figure 1G-H*). We also verified that the dependency of H in BOLD on $g$ was largely independent of the details of how BOLD is simulated from LFP. While the results shown in *Figure 2* are computed with an HRF that reproduces the correlation function measured between the BOLD signal and the gamma band of LFP (*Magri et al., 2012*), it is notable that these results remained similar when using the canonical HRF instead (*Figure 2—figure supplement 2*). Removing the high pass filter from simulation of BOLD response did alter the relative values of correlation between LFP power and BOLD across frequency bands, making the BOLD response more in disagreement with experimental data, but did not change the relationship of decreasing H with decreasing $g$ (*Figure 2—figure supplement 2*), suggesting that our conclusions are robust to the details of the model of the LFP to BOLD relationship. In sum, the inferences from the recurrent network model suggest that H in LFP and BOLD data can be utilized as a marker to track changes in underlying synaptic E:I mechanisms.



**Figure 2.** Relationship between E:I ratio from the recurrent model and H measured in simulated BOLD response. Panel **A** shows time-lagged Pearson correlations between LFP power across a range of different frequencies and the BOLD signal. Panel **B** shows the correlation between LFP power and the BOLD signal in selected frequency bands. The following four LFP bands are considered: alpha (8–12 Hz), beta (15–30 Hz), gamma (40–100 Hz) and the total LFP power (0–100 Hz). The relationship between E:I ratio (g) and H in simulated BOLD is shown in panels **C** and **D** with two different firing rates of thalamic input (1.5 and 2 spikes/second).

The online version of this article includes the following figure supplement(s) for figure 2:

**Figure supplement 1.** Simulating BOLD signal from LFP data generated by the recurrent model.

**Figure supplement 2.** Simulated BOLD with and without high-pass filter and changes to the HRF.

## Modeling the effects of chemogenetic manipulations within the recurrent network model

We next investigated manipulations of parameters within the recurrent model that approximate the effects of empirical chemogenetic DREADD manipulations in neurons. These simulations are useful to both gain a better understanding of the empirical BOLD measures under DREADD manipulations presented in the next section, and to better characterize the specificity of the origin of changes in 1/f slopes and H with the E:I ratio. Given that as shown above, in our models changes of H in BOLD mirror those in LFPs, here we present changes in model LFP spectra when simulating these DREADD manipulations.

We first studied the specific effect of solely increasing excitation within the recurrent network. This can be achieved experimentally by using the drug clozapine-N-oxide (CNO) on the DREADD receptor hM3Dq to increase the excitability of excitatory cells only (*Alexander et al., 2009*). We

simulated this kind of increase of excitability of pyramidal cells in the recurrent network model by lowering their voltage threshold ($V_{th}$) for spike initiation. Progressively lowering $V_{th}$ from -52 to -53 mV resulted in more positive low-frequency and flatter high-frequency 1/f slopes (*Figure 3—figure supplement 1A*) and also caused decreases in H (*Figure 3A*). For H, increasing $V_{th}$ (i.e. decreasing excitability) from -52 to -51 mV resulted in little change in H. These results predict that specific increases of excitation, as in the application of the hM3Dq DREADD to enhance excitability of

**Figure 3.** Changes to H after chemogenetic DREADD manipulations to enhance excitability or silence excitatory and inhibitory neurons. Panel **A** shows how H changes after the voltage threshold for spike initiation in excitatory neurons ($V_{th}$) is reduced in the recurrent model, thereby enhancing excitability as would be achieved with hM3Dq DREADD manipulation. Panel **B** shows how H changes after decreasing the resting potential of excitatory and inhibitory neurons ($E_L$), as would be achieved with hM4Di DREADD manipulation. Panel **C** shows changes in H in BOLD from prefrontal cortex (PFC) after real hM3Dq DREADD manipulation in mice, while panel **D** shows changes in PFC BOLD H after hM4Di DREADD manipulation in mice. In panels **C** and **D**, individual gray lines indicate H for individual mice, while the colored lines indicate Baseline (pink), Transition (green), and Treatment (blue) periods of the experiment. During the Baseline period H is measured before the drug or SHAM injection is implemented. The Transition phase is the period after injection but before the period where the drug has maximal effect. The Treatment phase occurs when the drug begins to exert its maximal effect. The green star indicates a Condition*Time interaction (p<0.05) in the Transition phase, whereas the blue stars indicate a main effect of Condition within the Treatment phase (p < 0.005).

The online version of this article includes the following figure supplement(s) for figure 3:

**Figure supplement 1.** 1/f slope computed with the FOOOF algorithm after manipulations to the voltage threshold for spike initiation and resting potential for excitatory and inhibitory neurons.

**Figure supplement 2.** This figure shows how fractional amplitude of low frequency fluctuations (fALFF) changes in BOLD signal from prefrontal cortex (PFC) after real hM3Dq DREADD (*i.e.* DREADD Excitation; left panel) or after hM4Di DREADD (*i.e.* DREADD Silencing; right panel) in mice.

pyramidal neurons, should reduce steepness of the high-frequency slopes and lead to a decrease in H. These results also confirm our above findings that in recurrent networks in which excitatory and inhibitory neurons interact, increases in excitability are easier to detect from changes in 1/f slope or H than decreases in excitation.

To study whether the changes in 1/f slopes and H are specific to modulations in excitability of only excitatory neurons, we modeled the combined effect of silencing both excitatory and inhibitory neuronal populations. This silencing of both excitatory and inhibitory neurons can be obtained experimentally by application of the hM4Di DREADD (see next Section). In the recurrent network model, we simulated this silencing of both excitatory and inhibitory cells by decreasing the resting potential, $E_L$, in both excitatory and inhibitory neurons. Decreasing $E_L$ from the baseline value of -70 to -75 mV produced varied effects in 1/f slopes (*Figure 3—figure supplement 1B*) and resulted in a slight increase of H (*Figure 3B*). Note that a moderate increase in H with higher input (*Figure 3B*) was also found when comparing two very different levels of input. Given that a possible non-local action of hM4Di might lead to less excitatory input to the considered area coming from the silencing of nearby regions, this suggests that our conclusion should still hold even in the presence of some non-local DREADD effects. In general, the effects of simulating hM4Di DREADD were far less prominent than those reported above when simulating enhanced excitation specifically (*Figure 3A* and *Figure 3—figure supplement 1A*). These results predict overall a very small effect of the hM4Di DREADD on H and 1/f slopes. These results also imply that decreases in H are more likely to result from specific increases in excitation rather than from non-specific decreases of excitability across both excitatory and inhibitory neuronal populations.

## Changes in H in BOLD after chemogenetic manipulation to enhance excitability of excitatory neurons in mice

All of the results thus far report results from our in-silico model of recurrent neuronal networks and their readouts as simulated LFP or BOLD data. The in-silico modeling of BOLD data suggests that if E:I ratio is increased via enhanced excitability of excitatory neurons, then H should decrease. To empirically test this prediction in-vivo, we measured rsfMRI BOLD signal in prefrontal cortex (PFC) of mice under conditions where a chemogenetic manipulation (hM3Dq DREADD) (*Alexander et al., 2009*) is used to enhance excitability of pyramidal neurons. Here we used a sliding window analysis to assess dynamic changes in H over the course of 3 different phases of the experiment – 1) a 'Baseline' phase where the CNO drug or a SHAM injection had not yet occurred, 2) a 'Transition' phase directly following CNO or SHAM injection, and 3) a 'Treatment' phase, whereby CNO has its maximal effect. We find that H is modulated over time by the DREADD manipulation (condition*time*treatment phase interaction $F = 349.03$, p<0.0001). During the Baseline phase of rsfMRI scanning before the DREADD-actuator CNO was injected, H under DREADD or a SHAM control conditions are not affected (condition main effect $F = 0.82$, p=0.37; condition*time interaction $F = 0.36$, p=0.54). However, during the Transition phase of the experiment where the CNO begins to have its effects, we find a condition*time interaction ($F = 4.94$, p=0.0262), whereby H drops over time at a steeper rate during the DREADD condition compared to the SHAM condition (green line in *Figure 3C*). Finally, during the Treatment phase of the experiment, where the drug exerts its maximal effect, there is a significant main effect of condition ($F = 12.92$, p=0.0011) and no condition*time interaction ($F = 0.66$, p=0.4182) (blue line in *Figure 3C*) (*Table 1*). This effect is explained by H being reduced in the DREADD vs SHAM condition. These in-vivo results are directly in line with the in-silico

**Table 1.** Results from DREADD excitation manipulation.

F-statistics (p-values in parentheses) for main effects of time, condition, and time*condition interaction for each of the 3 phases of the experiment (Baseline, Transition, Treatment). *=p < 0.05, **=p < 0.001.

|  | Time | Condition (DREADD - SHAM) | Time x Condition |
|---|---|---|---|
| Baseline | 0.82 (0.372) | 0.81 (0.369) | 0.36 (0.549) |
| Transition | 5.65 (**0.017**)* | 3.25 (0.081) | 4.94 (**0.026**)* |
| Treatment | 0.61 (0.433) | 12.92 (**0.001**)** | 0.66 (0.418) |

prediction that enhancing E:I ratio via enhancing the excitability of excitatory neurons results in a decrease in H (i.e. *Figure 1F–H*, *Figure 2C–D*, and *Figure 3A*).

## Chemogenetically silencing both excitatory and inhibitory neurons has no effect on H in BOLD

While the above results show that specific enhancement of excitability in excitatory neurons results in a decrease in BOLD H, it is an important negative control contrast to investigate whether non-specifically reducing the excitability of both excitatory and inhibitory neuronal populations might also affect H. This is an important negative control since if H were to change in a similar direction after this manipulation, it would make interpretations about decrease in H being due to increased E:I ratio via excitation problematic. The in-silico simulation of this manipulation (*Figure 3B*) would predict that H would not be changed much, and that if a change in H were to occur, it would be a slight increase rather than a decrease in H. By expressing the inhibitory hM4Di DREADD (*Stachniak et al., 2014*) under the control of a pan-neuronal promoter, we chemogenetically silenced both excitatory and inhibitory neurons in PFC of mice and re-ran the same rsfMRI neuroimaging protocol as before. While a significant 3-way interaction between condition, time, and treatment phase was present ($F$ = 85.8, p<0.0001), there were no strong main effects of condition or condition*time interactions in any of the baseline, transition, or treatment phases of the experiment (see *Table 2* and *Figure 3D*). Overall, these results along with the recurrent model simulation of hM4Di DREADD (*Figure 3B*) bolster strength of the interpretation that enhanced excitation drives decreasing H in BOLD and that H in BOLD would not change appreciably in a situation such as pan-neuronal silencing of both excitatory and inhibitory neurons.

Consistent with the idea that heightened excitation leads to flattening of the 1/f slope and reductions in H, we also computed a measure of the fractional amplitude of low frequency fluctuations (fALFF) (*Zou et al., 2008*). Given the effect of flattening 1/f slope, we expected that fALFF would show reductions due to the DREADD excitation manipulation but would show no effect for the DREADD silencing manipulation. These expectations were confirmed, as DREADD excitation results in a large drop in fALFF, which shows a stark drop off midway through the transition phase and stays markedly lower throughout the treatment phase when the drug has its maximal effects. In contrast, similar effects do not occur for the DREADD silencing manipulation (see *Figure 3—figure supplement 2*).

## Autism-associated genes in excitatory neuronal cell types in the human brain are enriched for genes that are differentially expressed by androgen hormones

The in-silico and in-vivo animal model findings thus far suggest that excitation affects metrics computed on neural time-series data such as 1/f slope and H. Applied to the idea of sex-related heterogeneity in E:I imbalance in autism, these results make the prediction that excitatory neuronal cell types would be the central cell type affecting such neuroimaging phenotypes in a sex-specific manner. To test this hypothesis about sex-specific effects on excitatory neuronal cell types, we examined whether known autism-associated genes that affect excitatory neuronal cell types (*Satterstrom et al., 2020*; *Velmeshev et al., 2019*) are highly overlapping with differentially expressed genes in human neuronal stem cells when treated with a potent androgen hormone, dihydrotestosterone (DHT) (*Lombardo et al., 2018b*; *Quartier et al., 2018*). Genes differentially

**Table 2.** Results from DREADD silencing manipulation.

F-statistics (p-values in parentheses) for main effects of time, condition, and time*condition interaction for each of the 3 phases of the experiment (Baseline, Transition, Treatment). *=p < 0.05, **=p < 0.001.

|  | Time | Condition (DREADD - SHAM) | Time x Condition |
|---|---|---|---|
| Baseline | 0.02 (0.876) | 4.01 (0.054) | 1.02 (0.137) |
| Transition | 0.04 (0.838) | 0.35 (0.561) | 1.36 (0.243) |
| Treatment | 0.40 (0.533) | 0.20 (0.673) | 0.10 (0.786) |

expressed by DHT are highly prominent within the gene set of autism-associated genes that affect excitatory neurons (OR = 1.67, p=0.03), with most of the overlapping genes being those whereby DHT upregulates expression (*Figure 4A*). By contrast, genes associated with autism that affect inhibitory neuronal cell types or other non-neuronal cells (e.g., microglia, astrocytes, oligodendrocytes) are not enriched for DHT differentially expressed genes (inhibitory neurons: OR = 1.51, p=0.12; microglia: OR = 0.78, p=0.78; astrocytes or oligodendrocytes: OR = 1.11, p=0.49). This result suggests that autism-associated genes specifically affecting excitatory neuronal cell types are also susceptible to the male-specific influence of androgen hormones in human neuronal stem cells.

We next additionally examined how such DHT-sensitive and autism-associated excitatory neuron genes spatially express in the adult human brain. This analysis would help shed insight on which brain areas might be more affected by such sex-specific effects in autism. A one-sample t-test of gene maps from the Allen Institute Human Brain Atlas (*Hawrylycz et al., 2012*) shows that this subset of DHT-sensitive and autism-associated excitatory neuron genes are highly expressed in MPFC, PCC, insula, and intraparietal sulcus, amongst other areas (*Figure 4B–C*).

## H is on-average reduced in adult autistic men but not women

We next move to application of this work to human rsfMRI data in autistic men and women. If E:I ratio is affected by sex-related mechanisms (*Lombardo et al., 2018b*), we predict that H would be differentially affected in autistic males versus females and manifest as a sex-by-diagnosis interaction

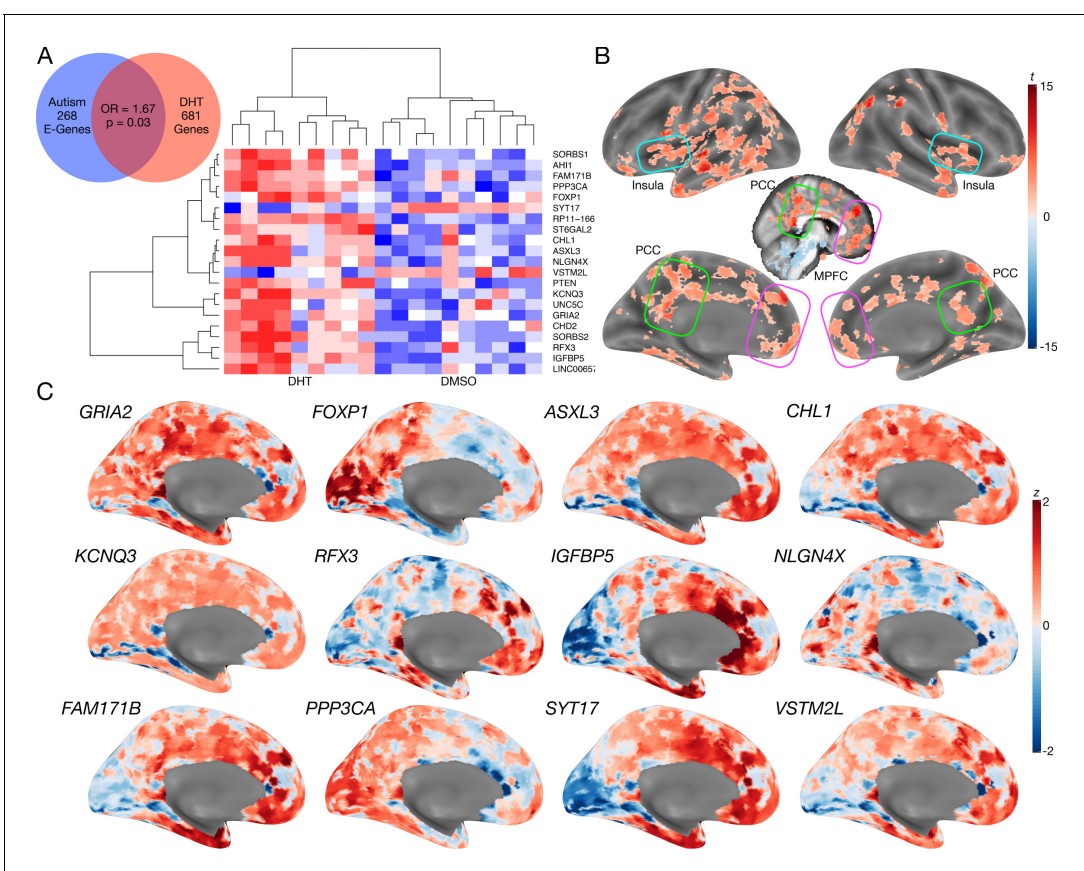

**Figure 4.** Autism-associated genes within excitatory neuronal cell types are enriched for genes differentially expressed by androgen hormones. Panel **A** shows a Venn diagram depicting the enrichment between autism-associated genes affecting excitatory neurons (Autism E-Genes) and DHT-sensitive genes. Panel **A** also includes a heatmap of these genes whereby the color indicates z-normalized expression values. The column dendrogram clearly shows that all samples with DHT treatment are clustered separately from the control (DMSO) samples. Each row depicts the expression of a different gene. Panel **B** shows a t-statistic map from a whole-brain one-sample t-test on these DHT-sensitive and autism-associated genes in excitatory neurons. Results are thresholded at FDR q < 0.01. Panel **C** shows spatial gene expression profiles on a representative surface rendering of the medial wall of the cortex for specific genes shown in panel B. Each map shows expression as z-scores with the color scaling set to a range of −2 < z < 2.

in a 2 × 2 factorial design (Sex: Male vs Female; Diagnosis: Autism vs Typically-Developing (TD)). More specifically, the directionality of our predictions from the in-silico and in-vivo results in *Figures 1–3* are that if H reflects E:I ratio, there should be decreased H (due to enhanced E) specifically in autistic males but not autistic females. Mass-univariate analysis uncovered one region in ventromedial prefrontal cortex (vMPFC), region p32, with a sex-by-diagnosis interaction passing FDR q < 0.05 ($F_{(5,104)}$ = 15.13, p=0.0001, *partial* $\eta^2$ = 0.12) (*Figure 5A*). In line with directionality of our predictions, this interaction effect is driven by a large TD >Autism effect in males (*Cohen's d* = 1.30) and a small Autism >TD effect in females (*Cohen's d* = −0.27) (*Figure 5B*). A similar sex-by-diagnosis interaction appeared when using another metric such as the intrinsic neural timescale (*Watanabe et al., 2019*; *Figure 5—figure supplement 1*) and when H was first calculated at each voxel and then

**Figure 5.** Autism rsfMRI sex-by-diagnosis interaction results. Panel **A** shows unthresholded and thresholded with FDR q < 0.05 mass-univariate results for the sex-by-diagnosis interaction contrast. Panel **B** shows H estimates from vMPFC (area p32) across males and females with and without autism. Panel **C** shows partial least squares (PLS) results unthresholded and thresholded to show the top 20% of brain regions ranked by bootstrap ratio (BSR). Panel **D** shows the percentage of voxels within each HCP-MMP parcellation region that overlap with the DHT-sensitive AND autism-associated genes affecting excitatory neurons (Autism E-Genes) map shown in *Figure 4B*. Panel **E** shows correlation between vMPFC H and behavioral camouflaging score in autistic males (orange) and females (blue).

The online version of this article includes the following figure supplement(s) for figure 5:

**Figure supplement 1.** Human rsfMRI data univariate main effects, PLS overlap with resting state networks, and effects on intrinsic neural timescale.

**Figure supplement 2.** This figure shows results from region p32 when H is computed first from all voxels and then averaged for a mean H estimate across all voxels from the p32 region.

averaged across voxels (*Figure 5—figure supplement 2*). While the main effects of diagnosis and sex are not the primary contrast for this study, we report that no significant regions survived FDR q < 0.05 for the main effects of diagnosis. However, 61% of brain regions showed an on-average male >female sex difference (*Figure 5—figure supplement 1*), which is in keeping with results from other work on sex differences in H (*Dhamala et al., 2020*).

In contrast to mass-univariate analysis, we also used partial least squares (PLS) analysis as a multivariate alternative to uncover distributed neural systems that express the sex-by-diagnosis interaction. PLS analysis identified one neural system expressing the same sex-by-diagnosis interaction (*d* = 2.04, p=0.036) and included default mode network (DMN) areas such as MPFC and posterior cingulate cortex/precuneus (PCC) (*Figure 5—figure supplement 1*), and other non-DMN areas such as insula, lateral prefrontal cortex, somatosensory and motor cortices, intraparietal sulcus, amongst others (*Figure 5C*). Many of these regions detected by the PLS analysis were subthreshold of FDR q < 0.05 in the mass-univariate analysis, but do show heightened effect sizes in keeping with this sex-by-diagnosis interaction pattern (e.g., white and light blue areas in the unthresholded map shown in *Figure 5A*). Detection of these regions in a mass-univariate analysis may require a larger sample size to enhance statistical power. Given that many of these PLS-identified regions of a sex-by-diagnosis interaction appear similar to those that appear in the gene expression map in *Figure 4B* of DHT-sensitive and autism-associated excitatory genes, we assessed how much each HCP-MMP parcellated regions overlap with the map in *Figure 4B*. PLS-identified regions in vMPFC (e.g., areas p32 and 10r) overlap by about 73–75%. Areas within the insula (e.g., Pol1, Pol2, MI) overlap by around 59–69%. Parietal areas in PCC (e.g., v23ab, d23ab) and intraparietal sulcus (LIPd) overlap by around 73–85% (*Figure 5D*).

## Correlation between vMPFC H and camouflaging in autistic women but not men

In prior task-fMRI work we found a similar sex-by-diagnosis interaction in vMPFC self-representation response and a female-specific brain-behavioral correlation with camouflaging ability (*Lai et al., 2019*). Given that adult autistic females engage more in camouflaging on-average (*Lai et al., 2017b*; *Hull et al., 2020*; *Schuck et al., 2019*), we next asked whether vMPFC H would be related to camouflaging in a sex-specific manner. In autistic females, increased camouflaging was strongly associated with increased H in vMPFC (*r* = 0.60, p=0.001). However, no significant association was apparent in autistic males (*r* = −0.10, p=0.63). The strength of this brain-behavioral correlation significantly differed between autistic males and females (*z* = 2.58, p=0.009) (*Figure 5E*). This result suggests that progressively more intact vMPFC H in autistic females, which are likely reflective of more intact E:I balance, is associated with better ability to camouflage social-communicative difficulties. Beyond this hypothesis-driven comparison of the relationship between H and camouflaging in vMPFC, we also ran correlations with ADI-R, ADOS and AQ scores. ADOS social-communication (SC) was negatively correlated with vMPFC H in autistic females (r = −0.51, p=0.008) indicating higher H with lower SC severity. This relationship was not present in autistic males (r = −0.04, p=0.83). However, the difference between these correlations was not statistically significant (z = 1.70, p=0.08). ADI-R subdomains, ADOS RRB, and AQ correlations were not statistically significant.

## Discussion

In this work we set out to better understand how intrinsic E:I imbalance affects the autistic brain in a sex-specific manner. Evidence from animal models of rare genetic variants associated with autism have typically been used as the primary evidence for the E:I imbalance theory (*Rubenstein and Merzenich, 2003*; *Sohal and Rubenstein, 2019*). However, these variants affect only a small percentage of the autism population. Thus, it is unclear how E:I imbalance might affect the majority of heterogeneous individuals within the total autism population. To bridge this gap we need multi-level methods that can be applied to understand the 'living biology' behind actual human individuals (*Courchesne et al., 2019*), such as in vivo neuroimaging data and metrics applied to such time-series data that are linked to actual underlying neural E:I mechanisms (*Markicevic et al., 2020*). Bridging this gap will help us identify mechanistic targets that explain neural and behavioral variability across a much larger portion of individuals in the autism population.

Based on earlier work (*Gao et al., 2017*), we reasoned that metrics such as 1/f slope and H in neural time-series data would be relevant as an in vivo neuroimaging marker of E:I mechanisms. Prior work suggested this relationship via a model that considers inhibition and excitation as separate entities (*Gao et al., 2017*). However, excitation and inhibition in the brain are inseparably linked. Results about the relationship between spectral shape and E:I balance obtained with our model of recurrent excitation and inhibition are largely compatible with those obtained with an earlier model of uncoupled excitation and inhibition (*Gao et al., 2017*). The uncoupled model predicts a linear increase of the slope value (i.e. flatter, less negative slopes) as E:I ratio increases. This is because in the uncoupled model changing the E:I ratio modifies only the ratio of the contribution to the LFP spectra of excitatory (faster time constant) and inhibitory (slower time constant) synaptic currents, leading to a linear relationship between slopes and E:I. In contrast, we found that the relationship between E:I and the spectral slope flattens out for high values of I. This, in our view, may in part arise from the fact that, as shown in studies of recurrent network models (*Brunel and Wang, 2003*), higher recurrent inhibition leads to higher peak frequency of gamma oscillations (i.e. an increase of power at higher frequencies) thus partly counteracting the low-pass filtering effect of inhibitory currents in the uncoupled model. We plan to investigate in future studies how these opposing effects interact in a wider range of configurations and to use these results to gain a better understanding of the relationship between E:I ratio and LFP spectral shape.

Furthermore, prior work (*Gao et al., 2017*) considered only 1/f slopes in simulated LFP data and did not explore the effect of the transformation between LFP neural activity to BOLD. Our simulations address these problems and significantly extends prior work (*Gao et al., 2017*) on the relationship between E:I imbalance and changes in spectral properties of neural signals. We showed that when excitation and inhibition interact in a recurrent network model, flatter 1/f slopes and decreases in H are specific markers of increases in E:I ratio. We also showed that in simulated BOLD signal, H and E:I ratio are associated in a manner similar to the relationships observed with LFP data. Taken together, these results predict that changes in H in neural time-series data can be interpreted as a shift in synaptic E:I ratio that permeates through in LFP or BOLD readouts.

Our simple model to generate BOLD from frequency-resolved LFPs reflect several features of the empirical LFP-BOLD relationship - namely the presence of a particularly strong gamma-BOLD relationship and the fact that a better prediction of the BOLD is obtained from the frequency-resolved LFP than from the wideband LFP. However, a limitation of our simple model is that, in its present form, it cannot capture the negative relationship between the power of some low-frequency LFP bands and the BOLD amplitude that has been reported in some studies (*Schölvinck et al., 2010*; *Scheeringa et al., 2011*; *Mukamel et al., 2005*; *Niessing et al., 2005*). Modelling the low frequency LFP to BOLD relationship in greater detail would require significant extensions of our neural model, as lower frequency oscillations are thought to arise from more complex cortico-cortical and thalamo-cortical loops than those that can be captured by our simple model of a local recurrent circuit with only two classes of neurons and no spatial structure (*Scheeringa and Fries, 2019*; *Zucca et al., 2019*). An important topic for further modelling work will be to understand how biomarkers of more complex neural feedback loops can be extracted from LFP or BOLD spectral signatures.

The power of our in-silico modeling approach is that it provides explicit predictions of what to expect in real BOLD data when synaptic E:I imbalance occurs. Remarkably, these in silico predictions are confirmed in vivo with rsfMRI BOLD data in halothane-sedated mice after experimental chemogenetic manipulations that specifically enhance neural excitation. Intriguingly, and consistent with in-silico predictions, manipulations that silence both excitatory and inhibitory neuronal populations do not have a strong effect on H in BOLD. These results are in line with optogenetic studies showing that specifically enhancing excitation in MPFC seems to have the biggest effects on social behavior in mice (*Yizhar et al., 2011*). The present work clearly shows that enhancement of excitation results in measurable changes in BOLD readouts as decreases in H. This insight allows us to leverage H as an in-vivo rsfMRI biomarker that has strong relevance back to synaptic E:I imbalance. Future extensions of our research might involve refined modelling and the use of chemogenetic manipulations in awake conditions, hence minimizing the possible confounding contribution of anesthesia on baseline E:I balance.

With regards to how sex-related heterogeneity in E:I imbalance might manifest in autism, we utilized genomics data and found that autism-associated genes that affect excitatory neuronal cell types are enriched for genes that are differentially expressed by DHT in human neuronal stem cells.

This inference extends prior work implicating excitatory neuron cell types in autism-relevant biology (*Satterstrom et al., 2020*; *Velmeshev et al., 2019*; *Willsey et al., 2013*; *Parikshak et al., 2013*) by linking genomic mechanisms in these cell types to the male-specific influence of androgen hormones. Importantly, other cell types such as inhibitory neurons do not express autism-associated genes that are also influenced by DHT. Additionally, the DHT-sensitive and autism-associated excitatory genes tend to spatially express in the human adult brain in regions such as MPFC, PCC, insula, and intraparietal sulcus, which have been shown to be affected in autism across a range of task-related and rsfMRI studies (*Lai et al., 2019*; *Di Martino et al., 2009*; *Di Martino et al., 2014*; *Uddin and Menon, 2009*; *Padmanabhan et al., 2017*; *Lombardo et al., 2010*), and which overlap with areas discovered by the PLS analysis to express a sex-by-diagnosis interaction (*Figure 5D*).

Moving to human rsfMRI data on adult individuals with autism, we utilized H as a neuroimaging biomarker of E:I imbalance. Specifically, we examined whether H differs between adult males and females with and without autism. Mass-univariate analysis highlighted one region in vMPFC which showed a sex-by-diagnosis interaction - that is, H was specifically reduced in adult autistic males, but not in autistic females. Reduced H in autistic males is compatible with the inference of elevated E:I ratio potentially driven by enhanced excitation. The observed effect in vMPFC may also be consistent with a 'gender-incoherence' pattern (i.e. towards reversal of typical sex differences in autism) (*Bejerot et al., 2012*). However, sex-specific normative ranges would need to be better established before interpreting effects in autism as being reversals of normative sex differences. More work with much larger general population-based datasets is needed to establish whether there are robust normative sex differences in H and to describe the normative ranges of H may take for each brain region, sex, and across age. Such work would also help with normative modeling (*Bethlehem and Seidlitz, 2018*) approaches that would enable identification of which autistic individuals highly deviate from sex-specific norms.

Multivariate PLS analysis extended the mass-univariate results by showing that a distributed neural system structurally and functionally connected to vMPFC, such as default mode network (DMN) areas like PCC (*Buckner and DiNicola, 2019*; *Yeo et al., 2011*), as well as intraparietal sulcus and insular cortex (*Uddin and Menon, 2009*), also expressed a similar but more subtle sex-by-diagnosis interaction. Interestingly, these regions highlighted by the PLS analysis are remarkably similar to the map of brain regions where autism-associated excitatory and DHT-sensitive genes highly express (*Figure 4B–C*, *Figure 5D*). Therefore, important social brain circuitry such as the DMN, and other integrative hubs of the salience network (e.g., insula) that connect DMN to other important large-scale networks (*Uddin and Menon, 2009*) may be asymmetrically affected by heightened E:I ratio in autistic males more than autistic females.

These human rsfMRI results are not only compatible with the in silico predictions and the in vivo mouse rsfMRI data presented here, but are also compatible with several prior lines of work. Our prior work highlighted that DMN functional connectivity in typically developing adolescent males, but not females, is affected by heightened levels of fetal testosterone and this network was heavily comprised of MPFC and PCC (*Lombardo et al., 2018b*). In the same work, we showed that a cortical midline DMN subsystem comprising MPFC and PCC highly expresses several genes relevant for excitatory postsynaptic potentials (e.g., *MEF2C, GRIK2, GRIA1, SCN3A, SCN9A, NPTX2*). The current findings linking autism-associated genes in excitatory neuron cell types (*Figure 4*) allow for more precise inferences about the importance of excitatory cell types over and above other inhibitory cell types. This is important given that evidence regarding inhibitory neuronal cell types and their role in E:I imbalance in autism is more mixed (*Horder et al., 2018*; *Coghlan et al., 2012*). Importantly, the expression of these genes in human neuronal stem cells is elevated after exposure to the potent androgen DHT (*Lombardo et al., 2018b*). Thus, one potential explanation for the male-specific reduction of H in vMPFC could have to do with early developmental and androgen-sensitive upregulation of genes that play central roles in excitatory neuron cell types, and thus ultimately affecting downstream E:I imbalance. Such effects may be less critical in human females and may serve an important basis for sex-differential human brain development (*Kaczkurkin et al., 2019*). These effects may also help explain why qualitative sex differences emerge in autism (*Lai et al., 2017a*; *Bedford et al., 2020*). rsfMRI H in autistic adults was also relevant in a sex-specific manner to a clinical behavioral phenomenon known as 'camouflaging'. Camouflaging relates to a set of compensatory or masking strategies/mechanisms that allow individuals to cope with their social-communicative difficulties in everyday social situations (*Lai et al., 2017b*; *Hull et al., 2020*;

*Livingston et al., 2019*). It is known that cognitively able adult autistic females tend to engage in more camouflaging behavior than males (*Lai et al., 2017b*; *Hull et al., 2020*; *Schuck et al., 2019*) and the extent to which individual females engage in camouflaging is linked to vMPFC function (*Lai et al., 2019*). One of the most important known functions of vMPFC has to do with self-representation (*Lombardo et al., 2010*) and simulating others based on information about the self (*Mitchell et al., 2006*). In prior task-related fMRI work we found a similar sex-by-diagnosis interaction effect whereby males are more impaired in vMPFC self-representation response than their female autistic counterparts. Furthermore, increased magnitude of vMPFC self-representation neural response correlates with increased camouflaging ability, but only in adult autistic females (*Lai et al., 2019*). Strikingly, here we find a similar sex-by-diagnosis interaction effect in vMPFC H as well as a female-specific correlation with camouflaging - as vMPFC H increases, indicative of a more normative or intact level of E:I balance, camouflaging also increases. This converging set of results suggests that intrinsic mechanisms such as E:I balance may be atypical only in cognitively able autistic males at vMPFC. More intact E:I balance in the vMPFC of autistic females may enable better vMPFC-related function (e.g., self-representation) and thus potentially better enable these individuals to camouflage social-communicative difficulties and cope in social situations. Future work changing E:I balance in vMPFC may provide a useful avenue for ameliorating daily life social-communication adaptation and coping difficulties in autistic males and enable them to optimally engage in compensatory processes such as camouflaging to the similar extent as autistic females. It may also be fruitful to examine how intact E:I balance in vMPFC of females may be an expression of protective factors that are hypothesized to buffer risk for autism in females (*Robinson et al., 2013*; *Werling, 2016*).

This work may also be of broader relevance for investigating sex-specific E:I imbalance that affects other early-onset neurodevelopmental disorders with a similar male-bias as autism (*Rutter et al., 2003*). For instance, conditions like ADHD affect males more frequently than females and also show some similarities in affecting behavioral regulation and associated neural correlates (*Chantiluke et al., 2015*). Furthermore, gene sets associated with excitatory and inhibitory neurotransmitters are linked to hyperactivity/impulsivity severity in ADHD, suggesting that E:I-relevant mechanisms may be perturbed (*Naaijen et al., 2017*). It will be important for future work to test how specific sex-specific E:I imbalance is to autism versus other related sex-biased neurodevelopmental disorders. Similarly, future work should investigate how H may change over development. Prior work has shown that H and other related measures such as 1/f slope can change with normative and pathological aging in both rsfMRI and EEG data (*Maxim et al., 2005*; *Wink et al., 2006*; *Voytek et al., 2015*). Imperative to this work will be the establishment of age and sex-specific norms for H in much larger datasets. Age and sex-specific norms will enable more work to better uncover how these biomarkers may be affected in neurodevelopmental disorders or disorders relevant to neurodegeneration. Such work combined with normative modeling approaches (*Bethlehem and Seidlitz, 2018*) may help uncover how experiential and environmental effects further affect such metrics.

In conclusion, we show that spectral properties of neural time-series data, such as H and 1/f slope, can be utilized in neuroimaging readouts like LFP and BOLD as a biomarker for underlying E:I-relevant mechanisms. In silico predictions from simulated LFP and BOLD data were confirmed in vivo with rsfMRI BOLD data where excitation was enhanced through chemogenetic manipulation. Finally, in application to humans, we show that H in rsfMRI data is reduced in vMPFC and other DMN areas of adult autistic males, but not females. Reduced H is indicative of enhanced excitation and thus points to sex-specific dysregulation of E:I balance in social brain networks of autistic males. This male-specific dysregulation of E:I balance may be linked to sex-differential early developmental events such as androgen-upregulation of gene expression for genes that play important roles in excitatory neurons (*Lombardo et al., 2018b*). The intact levels of H in females may help facilitate elevated levels of compensation known as camouflaging to cope with daily social-communicative difficulties. This important female-specific brain-behavioral correlation may also be key for future interventions targeting E:I mechanisms and MPFC-related brain networks to enable better coping with daily social-communicative difficulties. More generally, this work extends the relevance of the E:I imbalance theory of autism beyond evidence from autism-associated rare genetic variants and specify a larger portion of the autism population whereby these E:I mechanisms may be of critical importance.

## Materials and methods

### Human participants

All procedures contributing to this work comply with the ethical standards of the relevant national and institutional committees on human experimentation and with the Helsinki Declaration of 1975, as revised in 2008. All human participants' informed consent was obtained in accord with procedures approved by the Suffolk Local Research Ethics Committee. Adult native English speakers (n = 136, age range = 18–49 years) with normal/corrected-to-normal vision participated: n = 33 typically developing (TD) males, n = 34 autistic males, n = 34 TD females and n = 34 autistic females (*Table 3*). They all reported cis-gender identity based on a single item inquiring their birth-assigned sex and another on their identified gender. Groups were not statistically different on age or full-scale IQ (FIQ) on the Wechsler Abbreviated Scales of Intelligence (WASI) (*Table 3*). Exclusion criteria for all participants included a history of or current psychotic disorders, substance-use disorders, severe head injury, genetic disorders associated with autism (e.g. fragile X syndrome and tuberous sclerosis), intellectual disability (i.e. Full-scale IQ (FIQ) < 70), or other medical conditions significantly affecting brain function (e.g. epilepsy).

The inclusion criterion for both male and female autistic participants was a formal clinical diagnosis of International Statistical Classification of Diseases and Related Health Problems 10th Revision (ICD-10) childhood autism or Asperger's syndrome, or Diagnostic and Statistical Manual of Mental Disorders (4th ed., text rev.; DSM-IV-TR) autistic disorder or Asperger's disorder, as assessed by a psychiatrist or clinical psychologist in the National Health Service, UK. Since all participants were adults, we further considered available information of developmental history to include only those with clinically evident childhood autistic symptoms, for example, from information collected using the Autism Diagnostic Interview–Revised (ADI-R) (*Lord et al., 1994*) where possible, or from the participants' clinical diagnosis letters shared with the research team to determine eligibility. We used

**Table 3.** Descriptive and inferential statistics for group comparisons of demographic and clinical variables.

Values in the columns for each group represent the mean and standard deviation (in parentheses). Values in the columns labeled Sex, Diagnosis, and Sex*Diagnosis indicate the F-statistic and p-value (in parentheses). Abbreviations: TD, Typically Developing; VIQ, verbal IQ; PIQ, performance IQ; FIQ, full-scale IQ; ADI-R, Autism Diagnostic Interview–Revised; ADOS, Autism Diagnostic Observation Schedule; RRB, Restricted Repetitive Behaviors; AQ, Autism Spectrum Quotient; RMET, Reading the Mind in the Eyes Test, FD: frame-wise displacement.

| | TD males (N = 29) | Autistic males (N = 23) | TD females (N = 33) | Autistic females (N = 25) | Sex | Diagnosis | Sex* diagnosis |
|---|---|---|---|---|---|---|---|
| Age | 28.00 (6.42) | 27.13 (7.14) | 26.99 (5.34) | 27.35 (6.79) | 0.14 (0.70) | 0.03 (0.85) | 0.25 (0.61) |
| VIQ | 110.62 (11.53) | 114.70 (13.04) | 120.30 (10.06) | 114.08 (12.79) | 5.33 (0.02) | 0.38 (0.53) | 5.18 (0.02) |
| PIQ | 120.00 (10.21) | 114.57 (15.70) | 117.39 (9.27) | 110.88 (17.43) | 1.49 (0.22) | 5.55 (0.02) | 0.04 (0.83) |
| FIQ | 116.97 (10.69) | 116.39 (14.15) | 121.45 (8.33) | 114.16 (13.82) | 0.48 (0.48) | 3.38 (0.06) | 2.24 (0.13) |
| Camouflaging Score | - | −0.16 (0.38) | - | 0.15 (0.34) | 9.06 (0.004) | - | - |
| AQ | 15.28 (6.99) | 32.70 (8.47) | 11.97 (4.93) | 38.44 (6.34) | 0.26 (0.61) | 300.59 (2.2e-16) | 12.48 (0.0006) |
| ADI-R Reciprocal-Social-Interaction | - | 17.26 (4.77) | - | 16.56 (4.52) | 0.27 (0.60) | - | - |
| ADI-R Communication | - | 14.83 (3.50) | - | 13.40 (3.96) | 1.73 (0.19) | - | - |
| ADI-R RRB | - | 5.17 (2.35) | - | 4.24 (1.61) | 2.61 (0.11) | - | - |
| ADOS Communication | - | 3.30 (1.74) | - | 1.24 (1.30) | 21.85 (2.59e-5) | - | - |
| ADOS Social | - | 5.48 (3.45) | - | 3.48 (3.06) | 4.52 (0.03) | - | - |
| ADOS RRB | - | 1.09 (1.12) | - | 4.30 (1.61) | 61.84 (6.32e-10) | - | - |
| ADOS Communication + Social Total | - | 8.83 (4.87) | - | 4.72 (4.09) | 10.07 (0.002) | - | - |
| RMET | 27.14 (3.59) | 20.83 (6.87) | 28.91 (2.35) | 22.84 (6.40) | 3.93 (0.04) | 42.30 (2.704e-9) | 0.01 (0.89) |
| Mean FD | 0.17 (0.05) | 0.20 (0.07) | 0.18 (0.06) | 0.04 (0.17) | 0.51 (0.47) | 1.77 (0.18) | 1.10 (0.29) |

this clinically based criterion for inclusion for the purpose of sampling autistic individuals currently diagnosed by specialists in mental health services in the daily practice and to align with best clinical practice as recommended by the UK National Institute for Health and Clinical Excellence (NICE) guideline (*Pilling et al., 2012*). For assessing levels of autism characteristics, we administered the Autism Spectrum Quotient (AQ) (*Baron-Cohen et al., 2001a*), module 4 of the Autism Diagnostic Observation Schedule (ADOS) (*Lord et al., 2000*), and ADI-R (*Lord et al., 1994*) where possible, before the fMRI session. Autistic male and female groups were not significantly different on any ADI-R subdomain scores or Reading the Mind in the Eyes Test (RMET) (*Baron-Cohen et al., 2001b*) performance (*Table 3*).

We further used criteria for inclusion based on characteristics about data quality (see next paragraphs for data preprocessing). In particular, we excluded participants where the number of volumes was not acquired due to scanner hardware issues (n = 1), the preprocessing pipeline could not adequately preprocess the data (e.g., bad registrations; n = 5). Participants were also excluded if their head motion exceed a mean framewise displacement (meanFD) (*Power et al., 2012*) of >0.4 mm (n = 8). For the remaining subjects we further visually inspected plots of framewise displacement (FD) and DVARS (*Power et al., 2012*) traces to determine whether the wavelet despiking step sufficiently attenuated artefact-related variability that would leave DVARS spikes. Here we made a qualitative and consensus judgement amongst authors (S.T. and M.V.L) to exclude individuals (n = 9) whereby there were numerous FD spikes above 0.5 mm or numerous DVARS spikes leftover after wavelet despiking was applied. Other exclusions included any VIQ or PIQ <70 (n = 1) and co-morbid agenesis of the corpus callosum (n = 1). The final sample sizes included in all further analyses was n = 29 TD males, n = 23 autistic males, n = 33 TD females, and n = 25 autistic females. The final groups used in all analyses did not statistically differ on age (diagnosis main effect: $F(3,106)$ = 0.03, p=0.85; sex main effect: $F(3,106)$ = 0.14, p=0.70; sex-by-diagnosis interaction: $F(3,106)$ = 0.25, p=0.61) or FIQ (diagnosis main effect: $F(3,106)$ = 3.38, p=0.07; sex main effect: $F(3,106)$ = 0.48, p=0.48; sex-by-diagnosis interaction: $F(3,106)$ = 2.24, p=0.13) (see *Table 3*).

## Human fMRI data acquisition

Imaging was performed on a 3T GE Signa Scanner at the Cambridge Magnetic Resonance Imaging and Spectroscopy Unit. Participants were asked to lie quietly in the scanner awake with eyes closed for 13 min and 39 s during sequential acquisition of 625 whole-brain T2*-weighted echo planar image volumes with the following parameters: relaxation time = 1302 ms; echo time = 30 ms; flip angle = 70°; matrix size = 64×64; field of view = 24 cm; 22 anterior commissure-posterior commissure aligned slices per image volume; 4 mm axial slice thickness; 1 mm slice gap. The first five time-points were discarded to allow for T2-stabilization. During analysis of the Hurst exponent (H) for BOLD time-series, due to the discrete wavelet transform using volumes in power of 2, only the first 512 volumes ($2^9$) were utilized. A high-resolution spoiled gradient anatomical image was acquired for each participant for registration purposes.

## Human fMRI data analysis

Preprocessing of the resting state data was split into two components; core preprocessing and denoising. Core preprocessing was implemented with AFNI (*Cox, 1996*) (http://afni.nimh.nih.gov/) using the tool speedypp.py (http://bit.ly/23u2vZp) (*Kundu et al., 2012*). This core preprocessing pipeline included the following steps: (i) slice acquisition correction using heptic (7th order) Lagrange polynomial interpolation; (ii) rigid-body head movement correction to the first frame of data, using quintic (5th order) polynomial interpolation to estimate the realignment parameters (3 displacements and three rotations); (iii) obliquity transform to the structural image; (iv) affine co-registration to the skull-stripped structural image using a gray matter mask; (v) nonlinear warping to MNI space (MNI152 template) with AFNI 3dQwarp; (vi) spatial smoothing (6 mm FWHM); and (vii) a within-run intensity normalization to a whole-brain median of 1000. Core preprocessing was followed by denoising steps to further remove motion-related and other artifacts. Denoising steps included: (viii) wavelet time series despiking ('wavelet denoising'); (ix) confound signal regression including the six motion parameters estimated in (ii), their first order temporal derivatives, and ventricular cerebrospinal fluid (CSF) signal (referred to as 13-parameter regression). The wavelet denoising method has been shown to mitigate substantial spatial and temporal heterogeneity in motion-related artifact

that manifests linearly or non-linearly and can do so without the need for data scrubbing (*Patel et al., 2014*). Data scrubbing (i.e. volume censoring) cannot be used in our time-series-based analyses here as such a procedure breaks up the temporal structure of the time-series in such a way that invalidates estimation of the Hurst exponent (H) that examine long-memory characteristics. Wavelet denoising is implemented with the Brain Wavelet toolbox (http://www.brainwavelet.org). The 13-parameter regression of motion and CSF signals was achieved using AFNI 3dBandpass with the –ort argument. To further characterize motion and its impact on the data, we computed FD and DVARS (*Power et al., 2012*). Between-group comparisons showed that all groups were similar with respect to head motion as measured by meanFD with no diagnosis ($F_{(3,106)}$ = 1.77, p=0.18) or sex ($F_{(3,106)}$ = 0.51, p=0.47) main effects or sex-by-diagnosis interaction ($F_{(3,106)}$ = 1.10, p=0.29). All groups showed average meanFD of less than 0.2 mm (see *Table 3*).

Mean time-series for each of the 180 parcels within the Human Connectome Project Multimodal Parcellation (HCP-MMP) (*Glasser et al., 2016*) were extracted from the final preprocessed data, to estimate H. The estimation of H utilizes a discrete wavelet transform and a model of the time-series as fractionally integrated processes (FIP) and is estimated using maximum likelihood estimation. This method utilizing the FIP model for estimating H differs from our prior work (*Lai et al., 2010*), which used a model of fractional Gaussian noise (fGn). fGn is one type of process subsumed under the FIP model. However, the fGn model has the limitation of assuming that the BOLD time-series is stationary and also limits the upper bound of H at 1. In practice, we have seen that the upper bound of H = 1 from the fGn model results in ceiling effects for many brain regions and subjects. Thus, to remove the assumption of stationarity and upper bound of H = 1, the FIP model offers more flexibility and potentially added sensitivity due to better estimation of between-subject variability when estimates are near or exceed H = 1. When H > 1 the time-series is considered non-stationary and has long memory characteristics (e.g., is fractal). H is computed using the *nonfractal* MATLAB toolbox written by one of the co-authors (WY) (https://github.com/wonsang/nonfractal). The specific function utilized is bfn_mfin_ml.m function with the 'filter' argument set to 'haar' and the 'ub' and 'lb' arguments set to [1.5,10] and [−0.5,0], respectively.

After H was estimated for each of the 180 HCP-MMP parcels, we used a general linear model to test for sex-by-diagnosis interactions as well as main effects of Sex and Diagnosis in H. These models also incorporated meanFD and FIQ as covariates of no interest. Multiple comparison correction was achieved using an FDR q < 0.05 threshold. Visualization of effect sizes for figures was achieved using the *ggseg* library in R (https://github.com/LCBC-UiO/ggseg).

In addition to mass-univariate analysis, we also utilized multivariate partial least squares (PLS) analysis (*Krishnan et al., 2011*) to highlight distributed neural systems that capture the effect of a sex-by-diagnosis interaction. This analysis was implemented with code from the *plsgui* MATLAB toolbox (http://www.rotman-baycrest.on.ca/pls/). A matrix with participants along the rows and all 180 HCP-MMP parcels along with columns was input as the primary neuroimaging matrix for PLS. We also inserted a vector describing the sex-by-diagnosis contrast as the matrix to relate to the neuroimaging matrix. This vector describing the sex-by-diagnosis interaction was computed by matrix multiplication of the contrast vector of [1, -1, -1, 1] to a design matrix that was set up with columns defining TD males, autism males, TD females, and autism females, respectively. The PLS analysis was run with 10,000 permutations to compute p-values for each latent-variable (LV) pair and 10,000 bootstrap resamples in order to compute bootstrap ratios (BSR) to identify brain regions of importance for each LV pair. To isolate specific brain regions of importance for a statistically significant LV, we selected the top 20th percentile of brain regions ranked by BSR.

Relationships between H and camouflaging were conducted within autistic males and females separately. Pearson's correlations were used to estimate the strength of the relationship and groups were compared on the strength of the relationship using Fisher's r-to-z transform as implemented with the paired.r function in the *psych* library in R.

## Behavioral index of camouflaging

Camouflaging (consciously or unconsciously compensating for and/or masking difficulties in social–interpersonal situations) was operationalized as prior work (*Lai et al., 2017b*; *Lai et al., 2019*): the discrepancy between extrinsic behavioral presentation in social–interpersonal contexts and the person's intrinsic status. We used both the AQ score and RMET correct score as reflecting intrinsic status (i.e. self-rated dispositional traits and performance-based socio-cognitive/mentalizing capability),

and the ADOS Social-Communication total score as reflecting extrinsic behavioral presentation. The three scores were first standardized ($S_{ADOS}$, $S_{AQ}$ and $S_{RMET}$) within our sample of autistic men and women by mean-centering (to the whole autism sample in this study) and scaling (i.e. divided by the maximum possible score of each) to generate uniformly scaled measures that can be arithmetically manipulated. The first estimate of camouflaging was quantified as the difference between self-rated autistic traits and extrinsic behaviors ($CF1 = S_{AQ} - S_{ADOS}$), and the second estimate between mentalizing ability and extrinsic behaviors ($CF2 = -S_{RMET} - S_{ADOS}$). Then, using principal component analysis, the first principal component score of CF1 and CF2 (accounting for 86% of the total variance) was taken as a single, parsimonious measure of camouflaging for all subsequent analyses. This method was utilized in order to be consistent with prior work which computed the camouflaging metric in an identical fashion (*Lai et al., 2017b*; *Lai et al., 2019*). This measure should be interpreted by relative values (i.e. higher scores indicate more camouflaging) rather than absolute values. This operationalization only allows for estimating camouflaging in autistic individuals in our cohort, as it partly derives from the ADOS score which was not available in TD participants. This approach remains informative, as qualitative studies suggest that camouflaging in autism can be different from similar phenomenon (e.g. impression management) in TD individuals (*Bargiela et al., 2016*; *Hull et al., 2017*).

## In vivo chemogenetic manipulation of excitation in mouse prefrontal cortex

All in vivo studies in mice were conducted in accordance with the Italian law (DL 116, 1992 Ministero della Sanità, Roma) and the recommendations in the Guide for the Care and Use of Laboratory Animals of the National Institutes of Health. Animal research protocols were also reviewed and consented to by the animal care committee of the Istituto Italiano di Tecnologia. The Italian Ministry of Health specifically approved the protocol of this study, authorization no. 852/17 to A.G. All surgical procedures were performed under anesthesia.

Six to eight week-old adult male C57Bl6/J mice (Jackson Laboratories; Bar Harbor, ME, USA) were anesthetized with isoflurane (isoflurane 4%) and head-fixed in a mouse stereotaxic apparatus (isoflurane 2%, Stoelting). Viral injections were performed with a Hamilton syringe mounted on Nanoliter Syringe Pump with controller (KD Scientific), at a speed of 0.05 µl/min, followed by a 5–10 min waiting period, to avoid backflow of viral solution and unspecific labeling. Viral suspensions were injected bilaterally in PFC using the following coordinates, expressed in millimeter from bregma: 1.7 from anterior to posterior, 0.3 lateral, −1.7 deep. The inhibitory DREADD hM4Di was transduced using an AAV8-hSyn-hM4D(Gi)-mCherry construct. Control animals were injected with a control AAV8-hSyn-GFP virus (www.addgene.com). These viral suspensions were injected using a 0.3 µL injection volume in n = 15 hM4Di DREADD and n = 19 SHAM mice, respectively. The excitatory DREADD hM3Dq was transduced using an AAV8-CamkII-hM3D(Gq)-mCherry construct. Control animals for this experiment were injected with a control AAV8-CamkII-GFP construct. This set of injection were carried out using a 1 µL injection volume in n = 17 hM3Dq DREADD and n = 19 SHAM mice, respectively. We waited at least 3 weeks to allow for maximal viral expression.

## Mouse rsfMRI data acquisition

The animal preparation protocol for mouse rsfMRI scanning was previously described in detail (*Bertero et al., 2018*). Briefly, mice were anesthetized with isoflurane (5% induction), intubated and artificially ventilated (2% maintenance). Then isoflurane was discontinued and substituted with halothane (0.75%), a sedative that preserves cerebral blood flow auto-regulation and neurovascular coupling (*Gozzi et al., 2007*). Functional data acquisition commenced 30 min after isoflurane cessation. CNO (2 mg/kg for hM4Di and 0.5 mg/kg for hM3Dq) was administered i.v. after 15 min from the beginning of the acquisition both in virally transduced animals and in sham mice.

## Mouse rsfMRI data analysis

Raw mouse rsfMRI data was preprocessed as described in previous work (*Gutierrez-Barragan et al., 2019*; *Liska et al., 2015*). Briefly, the initial 120 volumes of the time-series were removed to allow for T1 and gradient equilibration effects. Data were then despiked, motion corrected and spatially registered to a common reference template. Motion traces of head realignment parameters (three

translations + three rotations) and mean ventricular signal (corresponding to the averaged BOLD signal within a reference ventricular mask) were used as nuisance covariates and regressed out from each time course. All rsfMRI time-series also underwent band-pass filtering to a frequency window of 0.01–0.1 Hz and spatial smoothing with a full width at half maximum of 0.6 mm.

The experimental design of the study allowed for computation of H during time-windows in the rsfMRI scan before drug injection (i.e. 'Baseline'), a transition phase where the drug begins having its effect (i.e. 'Transition'), and a treatment phase when the drug is thought to have its optimal effect (i.e. 'Treatment'). Analysis of condition, treatment phase, time, and all interactions between such factors was achieved using a sliding window analysis. Each window was 512 volumes in length and the sliding step was 1 vol. H is computed at each window and results in an H time-series. The H time-series is used as the dependent variable in a linear mixed effect model (i.e. using the *lme* function within the *nlme* library in R) with fixed effects of condition, time, treatment phase, and all 2-way and 3-way interactions between such factors as well as a factor accounting for scan day. Random effects in the model included time within mouse as well as treatment phase within mouse, all modeled with random intercepts and slopes. This omnibus model was utilized to examine a 3-way interaction between condition, time, and treatment phase. If this interaction was present, we then split the data by the 3 levels of the treatment phase (e.g., Baseline, Transition, and Treatment), in order to examine the main effect of condition or the condition*time interaction. Plots of the data indicate each mouse (grey lines in *Figure 3*) as well as group trajectories for each phase, with all trajectories estimated with a generalized additive model smoother applied to individual mice and group trajectories.

## In silico recurrent network modeling of LFP and BOLD data

The recurrent network model we use represents a standard cortical circuit incorporating integrate-and-fire excitatory and inhibitory spiking neurons that interact through recurrent connections and receive external inputs (both a sensory driven thalamic input and a sensory unrelated intracortical input, see *Figure 1A*). The network structure and parameters of the recurrent network model are the same ones used in *Cavallari et al., 2014* with conductance-based synapses (for full details see *Cavallari et al., 2014*). The network is composed of 5000 neurons, of which 4000 are excitatory (i.e. they form AMPA-like excitatory synapses with other neurons) and 1000 inhibitory (forming GABA-like synapses). Neurons are randomly connected with a connection probability between each pair of neurons of 0.2. Both populations receive two different types of external Poisson inputs: a constant-rate thalamic input and an intracortical input generated by an Ornstein-Uhlenbeck (OU) process with zero mean. A description of the baseline reference parameters used in simulations is given in *Table 4*. The LFP is computed as the sum of absolute values of AMPA and GABA postsynaptic currents on excitatory cells (*Mazzoni et al., 2008*; *Mazzoni et al., 2015*). This simple estimation of LFPs was shown to capture more than 90% of variance of both experimental data recorded from cortical field potentials and of simulated data from a complex three-dimensional model of the dipoles generated by cortical neurons (*Mazzoni et al., 2008*; *Mazzoni et al., 2015*; *Barbieri et al., 2014*). We changed the E:I ratio by independently varying the strengths of the inhibitory ($g_I$) and excitatory ($g_E$) synaptic conductances. We called $g$ the relative ratio between inhibitory and excitatory conductances ($g = g_I/g_E$). We present results of simulations for two levels of strength of thalamic input ($v_0$ = 1.5 spikes/second and $v_0$ = 2 spikes/second), and we verified that our results hold qualitatively for a wider range of input levels (1.5 to 4 spikes/second).

For the simulations used to compute H and 1/f slope of LFPs, we simulated a 10 s stretch of network activity from which we extracted a 10 s LFP time series used to compute H and 1/f slopes for each individual value of g (*Figure 1B*). To estimate power spectral density (PSD) we computed the Fast Fourier Transform with the Welch's method, dividing the data into ten overlapping segments with 50% overlap. 1/f slopes were computed with least-squares regressions predicting log power with log frequency. A piece-wise regression was applied to fit two line segments to the PSD – one segment to a low frequency region from 1 to 30 Hz and a second segment to a high-frequency region from 30 to 100 Hz.

As a basis for our model translating BOLD from LFP data, we note that prior studies with simultaneous electrophysiological and fMRI recordings in non-human primates have established that BOLD signal amplitude is more closely correlated with LFP than with any other type of neuronal events, such as spikes (*Logothetis et al., 2001*; *Magri et al., 2012*). Similarly, simultaneous

**Table 4.** Baseline reference parameters of the recurrent network model.
Parameters used in *Cavallari et al., 2014* with conductance-based synapses.

A: Neuron model

| Parameter | Description | Excitatory cells | Inhibitory cells |
|---|---|---|---|
| $V_{leak}$ (mV) | Leak membrane potential | −70 | −70 |
| $V_{threshold}$ (mV) | Spike threshold | −52 | −52 |
| $V_{reset}$ (mV) | Reset potential | −59 | −59 |
| $\tau_{refractory}$ (ms) | Absolute refractory period | 2 | 1 |
| $g_{leak}$ (nS) | Leak membrane conductance | 25 | 20 |
| $C_m$ (pF) | Membrane capacitance | 500 | 200 |
| $\tau_m$ (ms) | Membrane time constant | 20 | 10 |

B: Connection parameters

| Parameter | Description | Excitatory cells | Inhibitory cells |
|---|---|---|---|
| $E_{AMPA}$ (mV) | AMPA reversal potential | 0 | 0 |
| $E_{GABA}$ (mV) | GABA reversal potential | −80 | −80 |
| $\tau_{r(AMPA)}$ (ms) | Conductance rise time (AMPA) | 0.4 | 0.2 |
| $\tau_{d(AMPA)}$ (ms) | Conductance decay time (AMPA) | 2 | 1 |
| $\tau_{r(GABA)}$ (ms) | Conductance rise time (GABA) | 0.25 | 0.25 |
| $\tau_{d(GABA)}$ (ms) | Conductance decay time (GABA) | 5 | 5 |
| $\tau_l$ (ms) | Synapse latency | 0 | 0 |
| $g_{AMPA(rec.)}$ (nS) | AMPA conductance (recurrent) | 0.178 | 0.233 |
| $g_{AMPA(tha.)}$ (nS) | AMPA conductance (thalamic) | 0.234 | 0.317 |
| $g_{AMPA(cort.)}$ (nS) | AMPA conductance (intracortical) | 0.187 | 0.254 |
| $g_{GABA}$ (nS) | GABA conductance | 2.01 | 2.7 |

electroencephalogram (EEG)/fMRI studies in humans have found that the BOLD correlates with the EEG (*Scheeringa et al., 2011*; *Scheeringa et al., 2009*), which in turns correlates strongly with the LFP (*Whittingstall and Logothetis, 2009*). Importantly, the BOLD amplitude at any given time has been found to correlate preferentially with the power of high frequency bands. In particular, the BOLD amplitude correlates strongly with the gamma (40–100 Hz) band. However, the power distribution across frequency bands carries complementary information about the BOLD signal, meaning that each band contributes to the prediction of BOLD and predicting the BOLD signal directly from a wide band (i.e. the whole LFP spectrum) leads to poorer predictions of BOLD (*Logothetis et al., 2001*; *Magri et al., 2012*; *Schölvinck et al., 2010*; *Scheeringa et al., 2011*; *Goense and Logothetis, 2008*; *Kilner et al., 2005*). To account for these empirical observations, we have developed a model of BOLD signal that integrates contributions from different bands with a preferential contribution from high frequency bands.

To compute the simulated BOLD through the convolutions of the simulated LFPs, we needed to generate longer time-series than the initial 10 seconds simulated for LFPs. However, it was unfeasible to simulate very long BOLD time-series due to limitations on computational resources. We thus created, from the LFP data used for evaluations of individual g values, aggregated LFP time-series corresponding to different intervals of g (rather than individual values of g), as follows. The set of 10-second LFP time-series was divided into 3 equi-populated groups of $g$: $g < 7.5$, $7.5 < g < 11$ and $g > 11$. A concatenated LFP time-series was created for each group by randomly concatenating 20 LFP traces, which provided 200-second LFP signals. Low frequencies of the concatenated data were log-log linearly extrapolated based on the low-frequency slopes obtained in LFP log-log linear piecewise fitting. To account for statistical variability, the process of concatenating data was repeated 20 times for each group of g, randomly changing the order in which the individual LFP traces were combined within the group.

Once concatenated LFP time-series data was simulated, we compute the simulated BOLD time-series as the LFP data convolved not only with a hemodynamic response function (HRF), as in standard network models (*Deco et al., 2004*; *Wijeakumar et al., 2017*; *Buss et al., 2014*), but also with a high-pass filter (HPF) that gives more predictive power to higher LFP frequencies (*Figure 2—figure supplement 1B*). We have tested different parameters of the HPF, checking that changing the parameters produce qualitatively similar results and a monotonic correspondence between H of simulated LFP and H of simulated BOLD, and we opted for a HPF with a cutoff frequency (the frequency where the response is lowered by 3 dB) of 12.5 Hz and with a peak response at 20 Hz. The effect of the HPF was to attenuate low frequencies of the BOLD power distribution, partly compensating the low-frequency enhancement of HRFs, and to shift the peak frequency of BOLD power to 0.03 Hz, a value much closer to the peak frequency found in our real BOLD data and in most BOLD studies (*Alcauter et al., 2015*; *Allen et al., 2011*) with respect to the one that would have been obtained without convolution (see *Figure 2—figure supplement 1*).

To simulate BOLD response from the LFP data generated from the recurrent model, within the frequency domain we multiplied the LFP spectrum with spectra of the high-pass filter (HPF) and the hemodynamic response function (HRF), as follows:

$$FFT(BOLD) = FFT(LFP)FFT(HPF)[FFT(HRF) + \eta]$$

Where $FFT$ is the fast Fourier transform operation and $\eta$ is a constant white noise term. This noise term summarizes neurovascular relationship at frequencies not observable because they are faster than the BOLD acquisition frequencies. We assumed that the amplitude of $\eta$ was very small and we assigned small values to this noise term. We checked that the exact value of amplitude of $\eta$, or the spectral profile of this noise term (for example, simulating 1/f noise instead of white noise), did not alter the monotonic relationship between the simulated H of BOLD and LFP. Finally, the simulated BOLD time-series data was produced by applying the inverse Fourier transform of $FFT(BOLD)$ and then downsampling the resulting signal to a lower sampling rate, similar to that used in BOLD experiments (e.g., 0.5 Hz). The HRF used for simulating BOLD was the HRF from *Magri et al., 2012*, but similar results were obtained when using a canonical HRF (see *Figure 2—figure supplement 2*).

## Analyses examining enrichment of autism-associated genes in different cell types with genes differentially expressed by androgen hormones

To test hypotheses regarding cell types that may be affected by androgen influence, we examined genes linked to autism via rare de novo protein truncating variants that are enriched for expression in specific cell types (*Satterstrom et al., 2020*). Of the 102 genes reported by Satterstrom et al., we split these lists by enrichments in early excitatory neurons (C3), MGE derived cortical interneurons (C16), microglia (C19), and astrocytes or oligodendrocyte precursor cells (C4). In addition to high risk mutations linked to autism, we additionally used a list of genes differentially expressed (DE) in different cell types within post-mortem prefrontal and anterior cingulate cortex tissue of autistic patients (*Velmeshev et al., 2019*). These DE gene lists were split into cell types, and we examined DE genes in any excitatory neuronal cell class (L2/3, L4/L5/6), inhibitory cell classes (IN-PV, IN-SST, IN-VIP, IN-SV2C), microglia, astrocytes (AST-PP, AST-FB), and oligodendrocytes.

To test the question of whether cell type autism-associated gene lists were enriched for genes known to be differentially expressed by DHT, we used a previous DE gene list from an RNA-seq dataset of DHT administration to human neuronal stem cells (*Lombardo et al., 2018b*). Custom code was utilized to compute enrichment odds ratios and hypergeometric p-values for each enrichment test with different cell type autism-associated lists. The background total for these tests was the total number of genes considered in the original DHT-administration dataset (13,284).

To test how the DHT-sensitive and autism-associated genes in excitatory neurons are expressed across the human adult brain, we used whole-brain maps of expression for each gene in MNI space from the Allen Institute Human Brain Atlas (*Hawrylycz et al., 2012*). Maps for each gene were downloaded from the Neurosynth website (https://neurosynth.org/genes/) and then submitted to a one-sample t-test in SPM12, with a threshold of FDR q < 0.01.

## Data and code availability

Tidy data and analysis code are available at https://github.com/IIT-LAND/ei_hurst; *Trakoshis, 2020a*; copy archived at https://github.com/elifesciences-publications/ei_hurst. Source code of the recurrent network model is available at https://github.com/pablomc88/EEG_proxy_from_network_point_neurons; *Trakoshis, 2020b*; copy archived at https://github.com/elifesciences-publications/EEG_proxy_from_network_point_neurons. Raw RNA-seq data used to identify genes differentially expressed by DHT can be found in Gene Expression Omnibus (GSE86457). Data for the Allen Institute Human Brain Atlas can be found here: https://human.brain-map.org. Mapping of this data to MNI space can be found at the Neurosynth website (https://neurosynth.org/genes/).

## Acknowledgements

The Medical Research Council Autism Imaging Multicentre Study Consortium (MRC AIMS Consortium) is a UK collaboration between the Institute of Psychiatry, Psychology and Neuroscience (IoPPN) at King's College London, the Autism Research Centre, University of Cambridge and the Autism Research Group, University of Oxford.

## Additional information

### Competing interests

Edward T Bullmore: is employed half-time by the University of Cambridge and half-time at GlaxoSmithKline plc (GSK); he holds stock in GSK. All other authors have no conflict of interests to declare. The other authors declare that no competing interests exist.

### Funding

| Funder | Grant reference number | Author |
| --- | --- | --- |
| H2020 European Research Council | 755816 | Michael V Lombardo |
| Simons Foundation | 602849 | Stefano Panzeri |
| Medical Research Council | 400061 | Simon Baron-Cohen |
| H2020 European Research Council | 802371 | Alessandro Gozzi |

The funders had no role in study design, data collection and interpretation, or the decision to submit the work for publication.

### Author contributions

Stavros Trakoshis, Data curation, Formal analysis, Methodology, Writing - original draft, Writing - review and editing; Pablo Martínez-Cañada, Conceptualization, Data curation, Software, Formal analysis, Visualization, Methodology, Writing - original draft, Writing - review and editing; Federico Rocchi, Carola Canella, Data curation, Investigation, Methodology, Writing - original draft; Wonsang You, Software, Writing - original draft; Bhismadev Chakrabarti, Conceptualization, Investigation, Writing - original draft; Amber NV Ruigrok, MRC AIMS Consortium, Data curation, Investigation; Edward T Bullmore, Supervision, Writing - original draft; John Suckling, Conceptualization, Data curation, Supervision, Writing - original draft; Marija Markicevic, Valerio Zerbi, Conceptualization, Writing - original draft; Simon Baron-Cohen, Supervision, Funding acquisition, Writing - original draft, Project administration; Alessandro Gozzi, Conceptualization, Resources, Data curation, Supervision, Funding acquisition, Investigation, Methodology, Writing - original draft, Project administration, Writing - review and editing; Meng-Chuan Lai, Supervision, Funding acquisition, Writing - original draft, Writing - review and editing; Stefano Panzeri, Data curation, Software, Formal analysis, Supervision, Funding acquisition, Methodology, Writing - original draft, Writing - review and editing; Michael V Lombardo, Conceptualization, Data curation, Formal analysis, Supervision, Funding

acquisition, Investigation, Visualization, Methodology, Writing - original draft, Project administration, Writing - review and editing

## Author ORCIDs
Valerio Zerbi ⓘ http://orcid.org/0000-0001-7984-9565
Alessandro Gozzi ⓘ https://orcid.org/0000-0002-5731-4137
Stefano Panzeri ⓘ http://orcid.org/0000-0003-1700-8909
Michael V Lombardo ⓘ https://orcid.org/0000-0001-6780-8619

## Ethics
Human subjects: All procedures contributing to this work on human subjects comply with the ethical standards of the relevant national and institutional committees on human experimentation and with the Helsinki Declaration of 1975, as revised in 2008. All human participants' informed consent was obtained in accord with procedures approved by the Suffolk Local Research Ethics Committee.
Animal experimentation: All in-vivo studies in mice were conducted in accordance with the Italian law (DL 116, 1992 Ministero della Sanità, Roma) and the recommendations in the Guide for the Care and Use of Laboratory Animals of the National Institutes of Health. Animal research protocols were also reviewed and consented to by the animal care committee of the Istituto Italiano di Tecnologia. The Italian Ministry of Health specifically approved the protocol of this study, authorization no. 852/17 to A.G.

## Decision letter and Author response
Decision letter https://doi.org/10.7554/eLife.55684.sa1
Author response https://doi.org/10.7554/eLife.55684.sa2

# Additional files
## Supplementary files
• Supplementary file 1. Parameters for E:I model: Parameters utilized in the E:I model from *Gao et al., 2017*, for simulating LFP data.

• Transparent reporting form

## Data availability
Tidy data and analysis code are available at https://github.com/IIT-LAND/ei_hurst (copy archived at https://github.com/elifesciences-publications/ei_hurst). Source code of the recurrent network model is available at https://github.com/pablomc88/EEG_proxy_from_network_point_neurons (copy archived at https://github.com/elifesciences-publications/EEG_proxy_from_network_point_neurons). Raw RNA-seq data used to identify genes differentially expressed by DHT can be found in Gene Expression Omnibus (GSE86457). Data for the Allen Institute Human Brain Atlas can be found here: https://human.brain-map.org. Mapping of this data to MNI space can be found at the Neurosynth website (https://neurosynth.org/genes/).

The following previously published datasets were used:

| Author(s) | Year | Dataset title | Dataset URL | Database and Identifier |
|---|---|---|---|---|
| Quartier A, Jost B, Keime C, Piton A | 2018 | Genes and pathways regulated by androgens as possible contributors to the male excess observed in autism [RNA-Seq] | https://www.ncbi.nlm.nih.gov/geo/query/acc.cgi?acc=GSE86457 | NCBI Gene Expression Omnibus, GSE86457 |
| Hawrylycz MJ, Lein ES, Guillozet-Bongaarts AL, Shen EH, Ng L, Miller JA, Lagemaat LN, Smith KA, Ebbert A, Riley ZL, Abajian C, | 2012 | An anatomically comprehensive atlas of the adult human brain transcriptome | https://human.brain-map.org/static/download | Allen Human Brain Atlas, human.brain-map |

Beckmann CF, Bernard A, Bertagnolli D, Boe AF, Cartagena PM, Chakravarty MM, Chapin M, Chong J, Dalley RA, Daly BD, Dang C, Datta S, Dee N, Dolbeare TA, Faber V, Feng D, Fowler DR, Goldy J, Gregor BW, Haradon Z, Haynor DR, Hohmann JG, Horvath S, Howard RE, Jeromin A, Jochim JM, Kinnunen M, Lau C, Lazarz ET, Lee C, Lemon TA, Li L, Li Y, Morris JA, Overly CC, Parker PD, Parry SE, Reding M, Royall JJ, Schulkin J, Sequeira PA, Slaughterbeck CR, Smith SC, Sodt AJ, Sunkin SM, Swanson BE, Vawter MP, Williams D, Wohnoutka P, Zielke HR, Geschwind DH, Hof PR, Smith SM, Koch C, Grant SGN, Jones AR

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
