## [Decision Letter]

**Acceptance summary:**

This paper addresses circuit dysfunction in autism. In doing so, it nicely integrates computational (in silico) modelling with novel methods for the analysis of time series data, picking up on a recent thread on excitation-inhibition balance in neocortex. One very unique approach of the paper is the notion of a "behavioral camouflage" – that is, the ability mask social communicative difficulties through cognitive strategies: the authors find that a relatively intact E:I balance in medial prefrontal cortex may assist women with autism to recruit behavioural camouflage. This work obviously needs to be further nuanced with the various contributions of social gender roles, learning and cognitive strategies more broadly, as these pertain to persons with autism. This paper makes valuable contributions towards neurophysiological mechanisms of large-scale neurophysiological signals, the role of biophysical models in neuroscience, and our understanding of autism spectrum disorders.

**Decision letter after peer review:**

Thank you for submitting your article "Intrinsic excitation-inhibition imbalance affects medial prefrontal cortex differently in autistic men versus women" for consideration by *eLife*. Your article has been reviewed by three peer reviewers, and the evaluation has been overseen by a Reviewing Editor and Christian Büchel as the Senior Editor. The following individuals involved in review of your submission have agreed to reveal their identity: Takamitsu Watanabe (Reviewer #1); Richard Gao (Reviewer #2).

The reviewers have discussed the reviews with one another and the Reviewing Editor has drafted this decision to help you prepare a revised submission.

Summary:

This paper presents an impressive integration of in silico modelling, in-vivo rodent fMRI with chemogenetic manipulation and human fMRI. A number of key insights are advanced and integrated: the emerging and informative knowledge of time scale hierarchies in the brain; neurobiological and social aspects of autism spectrum disorder, including sex/gender considerations; and the role of computational models in linking micro- and macroscopic scales of investigation.

Revisions:

All three reviewers were impressed with the ambition and technical accuracy of the study. Their detailed technical suggestions are included below – all of which essentially request edits to the text (mainly Introduction and Discussion), more detailed appraisals of the data, and further visualizations. There are no new data acquisitions required and the additional analyses are suggested to improve the depth and quality of the presentation.

Reviewer 2 challenges the link between BOLD and LFPs versus spiking behaviour. Since the work of Logothetis and colleagues (and as confirmed subsequently), it is generally accepted that BOLD reflects broad spectrum LFP fluctuations more than spiking output, although this finding may be at times over-simplified. Therefore I suggest a brief discussion of this issue either in the Materials and methods (to justify the choice) or the Discussion (to acknowledge a partial limitation).

The reviewers' points are provided in full (because they are detailed and constructive), but where possible we will endeavour to rely on editorial discretion in appraising the revisions.

Reviewer #1:

In this work, the authors first demonstrated an association between E/I balance and Hurst component (H) by showing that [1] in a computational study, the Hurst component based on LFP was positively correlated with E/I balance, [2] also in-silico, the H based on BOLD signals was positively correlated with E/I balance –this part is predictable given prior work about the LFP-BOLD association – and [3] in vivo, the BOLD-based H was increased when neuronal excitability was enhanced.

Afterwards, they showed [4] MPFC was among human brain regions whose gene expression had both male-specific and autism-specific patterns, [5] in resting-state fMRI signals, a significant sex-diagnosis interaction effect was found in Hurst component in vMPFC and [6] only in the autistic female, the H was correlated with the camouflaging score.

Based on these observations, the authors claim that [A] Hurst component can be an index for the E/I balance and [B] the association between E/I balance and autism differs between sexes.

The theme (E/I balance, autism and sex difference) is important and each observation appears scientifically sound. In particular, the first half about the relationship between E/I balance and H (results [1]-[3] in above) is straight forward and looks solid. Relatively, the last half could be improved if the following concerns are addressed.

First, I think readers would like to know the spatial relationship between the MPFC found in the gene analysis and the vMPFC detected in the following rsfMRI analysis. In my understanding, if the authors want to say any link between sex-related genes and autism/sex-specific H changes, these two regions should be somewhat overlapped.

Second, I think readers would appreciate it if the authors supply some more information about associations between Hurst component in the vMFPC and behaviours/symptoms: at least, in both sexes, associations between H and symptom severity (total, social part and RRB component, respectively) in autistic individuals and those between H and AQ scores in both TD and autistic groups in both sexes. Such additional information would be quite helpful to re-interpret the function of the brain area in each sex and in TD and autism.

Reviewer #2:

In this study, Trakoshis, Martinez-Cañada and colleagues take a multi-scale approach to study how altered excitation-inhibition balance manifests in spectral features of the fMRI BOLD signal and apply it to study autism. Combining computational modeling of recurrent spiking neural networks, rodent chemogenetic experiments in-vivo, and sex- and autism-related gene expression in neuronal stem cells and post-mortem brain tissue, the authors relate changes in Hurst exponent (H) and 1/f exponent of the power spectrum to modulation of neuronal excitation specifically. With this link, they argue for a mechanistic interpretation of BOLD signal differences (in H) between autistic males and females, as well as the neural correlate of social camouflaging in females only.

The study is well-designed and logically presented, and I especially commend the multiscale approach that start with computational modeling to establish experimental hypotheses. Overall, I believe the paper is of quality and within the scope of *eLife*, as it integrates several levels of biology and thus appeals to a broad audience. Substantive concerns regarding modeling and analysis choices, as well as potentially problematic interpretations are listed below. I would recommend the manuscript for publication after those are addressed through additional analyses and writing changes, without collection of additional data.

Major comments:

1) I have two potential issues regarding the simulation of BOLD from LFP. First – and I do not claim domain expertise here whatsoever – it seems to me that BOLD should be simulated by convolving the HRF with the spiking output of the circuit, not the LFP. The LFP, as the authors note, is a combination of excitatory and inhibitory synaptic fluctuations. While taking the summation of the absolute value of those fluctuations will approximate local spiking, the latter is a true representation of the circuit's output, and is less affected by the strength of the inputs.

2) Second, and related, is that the authors draw attention to the frequency-dependent correlation between BOLD and LFP power, as correlation increases with frequency. But this pattern in correlation is essentially baked into the model itself, since a high-pass filter is applied after convolution with the HRF, imposing a higher correlation in higher frequencies. In addition, while this model partially recapitulates empirical data, it's also been shown that there is a negative correlation between LFP low-frequency power (e.g., α, 8-12 Hz) and BOLD (e.g., Mukamel et al., 2005). While it's good that the authors demonstrate changes in H to be unaltered by a conventional HRF-convolution model, I would like them to please comment on these two points, and whether the latter can be recapitulated in their model. If not, this is a good point of limitation to discuss.

3) When simulating the effect of DREADD silencing, do the neurons still spike at very low leak potentials? If not, then the LFP is effectively driven by the statistics of the cortical and thalamic inputs, which would represent a different circuit regime from the recurrent interactions enabled by higher leak potentials. While this does not change the conclusion of the study drastically, it would implicate the non-recurrent inputs to be the key factor in shaping H in this instance, which can alter the interpretation of the DREADD data.

4) A related point: the spiking network model represents some local circuit that receives cortical inputs. When simulating the effect of DREADD, only E_L_ is changed, but not the cortical inputs. How realistic is this assumption? In other words, how local is the effect of DREADD on the resting state activity in the PFC? Additionally, while the authors show changes in H (or the lack thereof) in the two DREADD experiments, does absolute amplitude of BOLD fluctuations change in the expected directions, e.g., an increase with excitation and decrease with silencing?

5) The rat is anesthetized prior to chemogenetic manipulation, which already shifts E:I balance. Without collecting data from pre-anesthesia periods, it's hard to say what those changes in H are, and how they compare relative to the DREADD manipulation. But the authors should at least discuss this point, and I would even suggest trying to simulate the additional effect of anesthesia to see if there is an interaction with DREADD (though this is not necessary).

6) E:I is interpreted to be normal in autistic females (no difference in H between TD and ASD females), but there is significant covariation with behavior, which is one of the main findings of the study. However, this begs the question: how much variation in H can be attributed to "healthy behavior-modulated variation", and how much to pathology? In Figure 5B, it looks like the range in H in the female autism cohort straddles the TD and ASD male cohorts, with comparable standard deviation. On the other hand, H does not correlate with the degree of social camouflaging in the male ASD cohort, yet there is still significant variation in the amount of camouflaging they are capable of. Is there a separate mechanism at play for camouflaging in males then? How do the authors settle these two seemingly contradictory findings?

7) Subsection “Human fMRI data analysis” appears to state that the mean of the BOLD timeseries across voxel is taken first, from which H is computed. Doesn't it make more sense to compute H for each voxel and then average after? How different are these two situations? I don't have an intuition for how averaging timeseries affect H, but an analogous situation in spectral analysis is destructive interference, where two high-amplitude oscillations may combine to a flat signal due to a phase difference.

8) Another method question: why not fit PLS directly with the 2-column matrix (CF1 and CF2) directly, since PLS is capable of outputting a single explanatory component, instead of performing PCA first. They should be similar?

9) Similarly (and maybe I'm not understanding something here), why not fit PLS or mass univariate over the entire cortex for camouflaging as well, as the authors did to find spatial specificity for the sex*diagnosis contrast, instead of only regressing in vMPFC specifically?

10) Lastly, while I appreciate the ample reference to Gao et al., 2017, I suggest the authors check out Lombardi et al., 2017 as well, as they also look at how 1/f changes as a result of E:I shifts, but with a recurrent and interacting-E:I spiking model that is arguably more similar to the model in this manuscript. However, their finding is in contrast to Gao et al., 2017, where 1/f flattens for increasing inhibition. This is worth referencing and noting in the discussions.

Reviewer #3:

This is an impressive piece of work combining multi-level methods and different modalities using both mouse and human data that substantially enhances our understanding of E:I imbalance in autism. The authors demonstrate and extend the link between E:I balance and the Hurst component and 1/f slope in local field potentials and rsfMRI data using a biologically more plausible network model than prior work. They further establish the spatial distribution of autism-linked genes associated with excitation and androgen-sensitive genes that coincide with the sex-differential region associated with the H-component in the human rsfMRI data. Interestingly, this region also shows a sex-differential relationship to camouflaging behavior. This is an impressive set of analyses with elaborate and thorough methodological steps. Listed below are some minor comments that can help increase the clarity of the manuscript and interpretation of the results.

1) The authors start the Introduction with the sentence that E:I imbalance affects many disorders. This calls for at least a short mention in the Discussion how findings are specific to autism.

2) The Introduction lacks a bit of a red line to my taste. It would be clearer to read if sub-paragraphs were linked to each other. The different sections make sense – however it is left to the reader to link them to each other. Given the complexity and many aspects touched on, I think the flow of the Introduction could still be improved a bit.

3) Could authors explain in more details what “in silico” refers to in this context? This might not be clear to a broad readership.

4) How was mean frame-wise displacement calculated? Power? Jenkinson?

5) In Table 3, could authors also report differences in symptom scores and camouflaging between autistic males and females.

6) What is the justification for the use of PLS on top? What additional information is revealed?

7) A large part of the results is in mouse data. The Introduction should thus also include a section introducing how E:I imbalance can be investigated in animal models. Right now, the Introduction refers solely to human participants, whereas the entire first Result section does not.

8) Given the results first format, it should be pointed out briefly which dataset was used for which set of results. The first section should clearly state that this is done in the mouse data for example.

9) The sex-differential results in humans are intriguing. However, when looking at Figure 5B, I wonder whether a gender-incoherent pattern is evident here? Could authors further discuss this? Also, in line with this, I wonder whether authors could refine this part of the Discussion stating: "Thus, one potential explanation for the male-specific reduction of H in vMPFC could have to do with early developmental and androgen-sensitive upregulation of genes that play central roles in excitatory neuron cell types, and thus ultimately affecting downstream E:I imbalance. Such effects may be sex-differential and thus less critical in human females, serving an important basis of sex-differential human brain development and explaining the sex-based heterogeneity and qualitative sex differences of autism neurobiology in human." Do authors refer to typical females here? The results look like TD females show a similar pattern to autistic males. So how do authors explain the differential pattern in TD females and autistic females.

10) I like the link to the camouflaging behavior. However, the question arises whether there is also a sex-differential link to core autistic symptoms such as for example repetitive behaviors that have also been shown to differ across autistic males and females?

11) Could authors include information on the age range of autistic subjects (currently, I can't see the age range anywhere) and discuss the potential effect of age on their results from a neurodevelopmental perspective?

---

## [Author Response]

Reviewer #1:[…]First, I think readers would like to know the spatial relationship between the MPFC found in the gene analysis and the vMPFC detected in the following rsfMRI analysis. In my understanding, if the authors want to say any link between sex-related genes and autism/sex-specific H changes, these two regions should be somewhat overlapped.

We have now calculated the percentage of voxels in each HCP parcel that overlap with the map shown in Figure 4B. Around 73% of region p32 of vMPFC overlaps with this map. Other areas detected in the PLS analysis are also highly overlapping such as areas of the insula, PCC, and intraparietal sulcus. This has been included as a panel in Figure 5.

“Given that many of these PLS-identified regions of a sex-by-diagnosis interaction appear similar to those that appear in the gene expression map in Figure 4B of DHT-sensitive and autism-associated excitatory genes, we assessed how much each HCP-MMP parcellated regions overlap with the map in Figure 4B. PLS-identified regions in vMPFC (e.g., areas p32 and 10r) overlap by about 73-75%. Areas within the insula (e.g., Pol1, Pol2, MI) overlap by around 59-69%. Parietal areas in PCC (e.g., v23ab, d23ab) and intraparietal sulcus (LIPd) overlap by around 73-85% (Figure 5D).”

Second, I think readers would appreciate it if the authors supply some more information about associations between Hurst component in the vMFPC and behaviours/symptoms: at least, in both sexes, associations between H and symptom severity (total, social part and RRB component, respectively) in autistic individuals and those between H and AQ scores in both TD and autistic groups in both sexes. Such additional information would be quite helpful to re-interpret the function of the brain area in each sex and in TD and autism.

We have now run analyses to look at correlations in ADOS and AQ. Correlations with region p32 are not present for any scale except ADOS SC in autism females (r = -0.51, p = 0.008), indicating higher H is related to lower SC severity. In autism males, this correlation is not significant (r = -0.04, p = 0.83). However, the difference between correlations in males versus females was not statistically significant (z = 1.70, p = 0.08). This result has now been inserted in the revised manuscript.

“Beyond this hypothesis-driven comparison of the relationship between H and camouflaging in vMPFC, we also ran correlations with ADI-R, ADOS and AQ scores. ADOS social-communication (SC) was negatively correlated with vMPFC H in autistic females (r = -0.51, p = 0.008) indicating higher H with lower SC severity. This relationship was not present in autistic males (r = -0.04, p = 0.83). However, the difference between these correlations was not statistically significant (z = 1.70, p = 0.08). ADI-R subdomains, ADOS RRB, and AQ correlations were not statistically significant.”

Reviewer #2:[…]Major comments:1) I have two potential issues regarding the simulation of BOLD from LFP. First – and I do not claim domain expertise here whatsoever – it seems to me that BOLD should be simulated by convolving the HRF with the spiking output of the circuit, not the LFP. The LFP, as the authors note, is a combination of excitatory and inhibitory synaptic fluctuations. While taking the summation of the absolute value of those fluctuations will approximate local spiking, the latter is a true representation of the circuit's output, and is less affected by the strength of the inputs.

The reviewer brings up two important questions in this comment. The first question pertains to whether BOLD correlates more with spiking activity (a measure of the circuit’s output) or with the LFP. The second question regards why we computed the LFP from network activity as a sum of absolute values of synaptic currents.

To answer the first question, and as the Editor points out in their summary notes of the revisions, this has been addressed extensively by the seminal and widely known work of Logothetis, as well as work of others, on simultaneous recordings of intracranial neural activity and BOLD (to which some of us have also contributed). A fair but a little oversimplified summary is that in these experiments it was consistently found that BOLD correlates with both LFPs and spikes, but that, when LFPs and BOLD are dissociated (by pharmacological techniques or within natural fluctuations of brain activity), BOLD correlates more with the LFP (Logothetis et al., 2001; Magri et al., 2012; Rauch et al., 2008; Viswanathan and Freeman, 2007; Lauritzen and Gold, 2003). These findings are usually interpreted as implying that the BOLD contrast mechanism reflects the synaptic input and intracortical processing of a given area rather than its spiking output. When considering frequency-resolved LFP activity, in general the BOLD signal correlates more strongly with the power of high frequency LFP bands. In the previous version of the paper, we covered in our text the result that BOLD correlates more with gamma band of LFP power than with other bands. However, we did not discuss that BOLD correlates more with the LFP than with spikes. We now added this information, and related citations, to the text of the revised manuscript.

“Studies with simultaneous LFP and BOLD measured in animals have shown that although BOLD signal correlates with both LFPs and spikes, it correlates more strongly with the LFP than with spikes (Logothetis et al., 2001; Magri et al., 2012; Rauch, Rainer and Logothetis, 2008; Viswanathan and Gold, 2003).”

The reviewer’s second question dealt with why we computed the LFP from network activity as a sum of absolute values of synaptic currents. Following established and well validated procedures, we computed the simulated LFP from the model network of point-like integrate-and-fire neurons as the sum of absolute values of synaptic currents because AMPA synapses are usually apical and GABA synapse are peri-somatic and thus their dipoles sum with the same sign along dendrites (Mazzoni et al., 2008; Mazzoni et al., 2010; Deco et al., 2004). We have extensively validated this method of computing LFPs from integrate-and-fire networks in previous work. For example, we reported in previous work that computing the LFP as a sum of absolute values of synaptic currents gives a much better fit to real cortical LFP data (Mazzoni et al., 2008) or to LFPs generated by anatomically detailed three-dimensional networks models (Mazzoni et al., 2015) than using alternatives such as the sum of simulated membrane potentials, the signed sum of synaptic currents or a time integration of the spike rate.

“From this model, we computed the network’s LFP as the sum of absolute values of all synaptic currents. The absolute value is taken because AMPA synapses are usually apical and GABA synapse are peri-somatic and thus their dipoles sum with the same sign along dendrites (Mazzoni et al., 2008; Mazzoni et al., 2010 and Deco, Rolls and Horwitz 2004). We computed LFP summing presynaptic currents from both external inputs and recurrent interactions, as real LFPs capture both sources of synaptic activity(Logothetis, 2008). We have extensively validated this method of computing LFPs from integrate-and-fire networks in previous work on both real cortical data and simulations with networks of realistically-shaped 3D neurons and shown that it works better than when using alternatives such as the sum of simulated membrane potentials, the signed sum of synaptic currents or a time integration of the spike rate (Mazzoni et al., 2008 and Mazzoni et al., 2015.”

2) Second, and related, is that the authors draw attention to the frequency-dependent correlation between BOLD and LFP power, as correlation increases with frequency. But this pattern in correlation is essentially baked into the model itself, since a high-pass filter is applied after convolution with the HRF, imposing a higher correlation in higher frequencies. In addition, while this model partially recapitulates empirical data, it's also been shown that there is a negative correlation between LFP low-frequency power (e.g., α, 8-12 Hz) and BOLD (e.g., Mukamel et al., 2005). While it's good that the authors demonstrate changes in H to be unaltered by a conventional HRF-convolution model, I would like them to please comment on these two points, and whether the latter can be recapitulated in their model. If not, this is a good point of limitation to discuss.

We do not think that the BOLD-LFP pattern of correlation is hard-wired into the model by construction. In Figure 2—figure supplement 2A (Magri’s et al., 2012 HRF) and Figure 2—figure supplement 2B (canonical HRF), we show that removing the high-pass filter from simulation of BOLD response does not modify significantly the pattern of correlation between LFP power at high frequency bands (including gamma) and BOLD. The high-pass filter only adds an improvement in the fitting of experimental data in terms of the relative distribution of values of correlation across frequency bands. We added a note to the main text to highlight this fact.

“Removing the high pass filter from simulation of BOLD response did alter the relative values of correlation between LFP power and BOLD across frequency bands, making the BOLD response more in disagreement with experimental data, but did not change the relationship of decreasing H with decreasing g (Figure 2—figure supplement 2), suggesting that our conclusions are robust to the details of the model of the LFP to BOLD relationship.”

We also thank the reviewer for pointing out the need to discuss the sometimes-observed negative correlation between the low-frequency LFP power and BOLD. While our simplified model to generate BOLD from LFPs captures several empirical features of the LFP-BOLD relationship (namely the presence of a particularly strong gamma power-BOLD relationship and the fact that a better prediction of the BOLD is obtained from the frequency-resolved LFP than from the wideband LFP), our model is not able to capture all experimentally reported relationships, including the low frequency LFP power-to-BOLD anticorrelation pointed out by the reviewer. We feel that modelling the low frequency LFP power-to-BOLD relationship in greater detail would be outside the scope of this work. It is thought, in fact, that lower frequency oscillations arise from more complex cortico-cortical and thalamocortical loops than those that can be captured by a simple model of a local recurrent circuit with only two classes of neurons and no spatial structure, as it is ours. Following the recommendations of the Editor in the summary, we addressed this point by pointing out this limitation of our work, and the need for further investigations, in Discussion.

“Our simple model to generate BOLD from frequency-resolved LFPs reflect several features of the empirical LFP-BOLD relationship – namely the presence of a particularly strong gamma-BOLD relationship and the fact that a better prediction of the BOLD is obtained from the frequency-resolved LFP than from the wideband LFP. However, a limitation of our simple model is that, in its present form, it cannot capture the negative relationship between the power of some low-frequency LFP bands and the BOLD amplitude that has been reported in some studies (Schölvinck et al., 2010; Scheeringa et al., 2011; Mukamel, 2005 and Niessing, 2005). Modelling the low frequency LFP to BOLD relationship in greater detail would require significant extensions of our neural model, as lower frequency oscillations are thought to arise from more complex cortico-cortical and thalamocortical loops than those that can be captured by our simple model of a local recurrent circuit with only two classes of neurons and no spatial structure (Scheeringa and Fries, 2019; Zucca et al., 2019). An important topic for further modelling work will be to understand how biomarkers of more complex neural feedback loops can be extracted from LFP or BOLD spectral signatures.”

3) When simulating the effect of DREADD silencing, do the neurons still spike at very low leak potentials? If not, then the LFP is effectively driven by the statistics of the cortical and thalamic inputs, which would represent a different circuit regime from the recurrent interactions enabled by higher leak potentials. While this does not change the conclusion of the study drastically, it would implicate the non-recurrent inputs to be the key factor in shaping H in this instance, which can alter the interpretation of the DREADD data.

In response to the reviewer’s question, in our simulations excitatory and inhibitory populations spike for all simulation scenarios reported in the paper, including the data points with the highest value of inhibition within an individual graph and in the DREADD silencing simulations mentioned by the reviewer. Of course, firing rates of E and I neurons are lower when increasing inhibition or lowering resting potentials. In both real cortical data and in recurrent network simulations, it is well known that the LFP is driven by presynaptic currents from both external inputs and recurrent interactions. Because of this, the fact that the BOLD follows LFPs and not spikes has been interpreted by Logothetis and colleagues to imply that the LFP reflects the total processing taking place in a cortical areas (including inputs and recurrent contributions) and not its output. We edited the text to better specify that the LFP spectral shapes, in both real data and our simulations, reflect presynaptic currents from both external inputs and recurrent interactions.

“From this model, we computed the network’s LFP as the sum of absolute values of all synaptic currents. The absolute value is taken because AMPA synapses are usually apical and GABA synapse are peri-somatic and thus their dipoles sum with the same sign along dendrites (Mazzoni et al., 2008; Mazzoni et al., 2010 and Deco, Rolls and Horwitz, 2004). We computed LFP summing presynaptic currents from both external inputs and recurrent interactions, as real LFPs capture both sources of synaptic activity (Logothetis, 2008). We have extensively validated this method of computing LFPs from integrate-and-fire networks in previous work on both real cortical data and simulations with networks of realistically-shaped 3D neurons and shown that it works better than when using alternatives such as the sum of simulated membrane potentials, the signed sum of synaptic currents or a time integration of the spike rate (Mazzoni et al., 2008; Mazzoni et al., 2015).”

4) A related point: the spiking network model represents some local circuit that receives cortical inputs. When simulating the effect of DREADD, only E_L_ is changed, but not the cortical inputs. How realistic is this assumption? In other words, how local is the effect of DREADD on the resting state activity in the PFC?

At the cellular level, the effect of DREADD CNO-mediated activation of hM4Di on silencing neurons has been reported as a decrease of their excitability and of spiking activity, at least in part due to G protein inwardly rectifying potassium (GIRK) channels (Armbruster et al., 2007; Atasoy et al., 2012; Carter et al., 2013; Ferguson et al., 2011; Krashes et al., 2011; Ray et al., 2011; Sasaki et al., 2011).

In the previous version of the paper we simulated the effect of DREADD silencing (hM4Di applied under the control of a pan-neural promoter) by simulating a decrease in excitability through decreasing the resting potential of all neurons (Figure 3B), finding a very moderate increase in H for lower values of the resting potential. Regarding locality of DREADD effect, it should be pointed out that DREADD stimulation is carried out via systemic CNO administration (Roth, 2016), and regional targeting is achieved via localized expression of recombinant adenocairal vectors, which in the case of the present study where transfected in the mouse prefrontal cortex. Given the anterograde nature of the vectors employed, a long range effect of our manipulations cannot be ruled out.

We thus agree with the reviewer that it is interesting to model in simulations the possible non-locality of DREADD silencing effect through a decrease in the excitatory input to the network (which in our model goes to both E and I neurons, according to established anatomical findings). In the revised paper, we address this question by reporting in the revised text that in the simulations of Figure 3B we compare two different levels of input to the network and we found higher H for lower values of input, thus suggesting that our conclusion should still hold even in the presence of some non-local DREADD effects.

“Note that a moderate increase in H with higher input (Figure 3B) was also found when comparing two very different levels of input. Given that a possible non-local action of hM4Di might lead to less excitatory input to the considered area coming from the silencing of nearby regions, this suggests that our conclusion should still hold even in the presence of some non-local DREADD effects.”

Additionally, while the authors show changes in H (or the lack thereof) in the two DREADD experiments, does absolute amplitude of BOLD fluctuations change in the expected directions, e.g., an increase with excitation and decrease with silencing?

In order to address this question, we computed fractional amplitude of low frequency fluctuations (fALFF) (Zuo et al., 2008) on PFC BOLD timeseries data from the DREADD experiments. Given that 1/f slope flattens and H is reduced due to heightened excitation, we expected that the amplitude of low frequency fluctuations would be lowered as well. Using the same type of sliding window analysis as reported for H in the manuscript, we show that fALFF indeed drops systematically as a result of the DREADD excitation manipulation. In contrast, no such effects exist for the DREADD silencing manipulation. Although there are condition effects with the baseline period, those effects are weak in nature and also differ in opposing directions across the two DREADD manipulations. Furthermore, the lower baseline level of fALFF in the DREADD excitation experiment is not sufficient to explain the much larger drop in fALFF which is obvious in the plots, given the steep drop off halfway through the transition phase and which stays consistently lower throughout the treatment phase when the drug has its maximal effects.

“Consistent with the idea that heightened excitation leads to flattening of the 1/f slope and reductions in H, we also computed a measure of the fractional amplitude of low frequency fluctuations (fALFF)(Zou et al., 2008). Given the effect of flattening 1/f slope, we expected that fALFF would show reductions due to the DREADD excitation manipulation but would show no effect for the DREADD silencing manipulation. These expectations were confirmed, as DREADD excitation results in a large drop in fALFF, which shows a stark drop off midway through the transition phase and stays markedly lower throughout the treatment phase when the drug has its maximal effects. In contrast, similar effects do not occur for the DREADD silencing manipulation (see Figure 3—figure supplement 2).”

5) The rat is anesthetized prior to chemogenetic manipulation, which already shifts E:I balance. Without collecting data from pre-anesthesia periods, it's hard to say what those changes in H are, and how they compare relative to the DREADD manipulation. But the authors should at least discuss this point, and I would even suggest trying to simulate the additional effect of anesthesia to see if there is an interaction with DREADD (though this is not necessary).

Because of the difficulty in controlling restraint-induced stress in highly rousable rodent species, our chemo-fMRI manipulations were carried out under light sedation. Our sedation protocol ensures physiologically stability, preserves sensorial responsivity (Orth et al., 2006) and does not induce burst-suppression or cortical hyper-synchronization (Liu et al., 2011), resulting in preserved rsfMRI connectivity topography (Coletta et al., 2020; Whitesell et al., 2020) and rich fMRI state dynamics (Gutierrez-Barragan et al., 2019). While it is conceivable that the E:I balance produced by the sedative agent could be brain-state dependent, and as such partly affected by arousal state, converging lines of investigation argue against a significant interaction between anesthesia and DREADD manipulations, supporting the notion that the results of our manipulations are meaningful and translationally relevant, independent of the specific E:I state that characterizes our chemo-fMRI recordings.

In addition, there are other considerations to note. First, all of our quantifications have been expressed with respect to a reference control arm in both the excitatory and inhibitory DREADD studies, an experimental option that permits to control and account for possible drifts in the physiological state across time. Second, consistent with a negligible DREADD/anesthesia interaction, the results of our in silico modelling, in which no explicit account of anesthesia was included, accurately predicted empirical BOLD rsfMRI recordings as well as dynamic timeseries features observed in awake conditions in autistic individuals. Finally, the successful identification of sex-specific features based on H mapping in autism is also consistent with this view, suggesting that our modelling, and its empirical validation, can capture relevant features that can be extended to the quite wakefulness that characterize rsfMRI recordings in human populations (Reimann and Niendorf, 2020).

While a formal modelling of the effect of anesthesia would be beyond the scope of the present work in the revised manuscript, we acknowledge the use of DREADD manipulations in sedated animals as a potential limitation of our study.

“Future extensions of our research might involve refined modelling and the use of chemogenetic manipulations in awake conditions, hence minimizing the possible confounding contribution of anesthesia on baseline E:I balance.”

6) E:I is interpreted to be normal in autistic females (no difference in H between TD and ASD females), but there is significant covariation with behavior, which is one of the main findings of the study. However, this begs the question: how much variation in H can be attributed to "healthy behavior-modulated variation", and how much to pathology? In Figure 5B, it looks like the range in H in the female autism cohort straddles the TD and ASD male cohorts, with comparable standard deviation.

The reviewer raises an important question regarding normative ranges of H in the human brain (and its association pattern with behaviors) and how these vary by factors such as sex. In the absence of large-scale data from the general population, it is difficult to judge what would be the range of healthy versus pathological variation here without better estimates of population-level norms (e.g., mean/median, range, variance) for each region and by sex, and, preferably across age. We would suggest that future work is needed on very large sample datasets from the general population to estimate such norms. Once there is better evidence of robust population-level norms, it may be clearer what could be considered the range of healthy variability versus pathology. We intend to assess this question in future work.

“The observed effect in vMPFC may also be consistent with a “gender-incoherence” pattern (i.e. towards reversal of typical sex differences in autism)(Bejerot et al., 2012). However, sex-specific normative ranges would need to be better established before interpreting effects in autism as being reversals of normative sex differences. More work with much larger general population-based datasets is needed to establish whether there are robust normative sex differences in H and to describe the normative ranges of H may take for each brain region, sex, and across age. Such work would also help with normative modelling (Bethlehem et al., 2018) approaches that would enable identification of which autistic individuals highly deviate from sex-specific norms.”

On the other hand, H does not correlate with the degree of social camouflaging in the male ASD cohort, yet there is still significant variation in the amount of camouflaging they are capable of. Is there a separate mechanism at play for camouflaging in males then? How do the authors settle these two seemingly contradictory findings?

Regarding this sex-differential correlation with camouflaging, it is indeed likely that different mechanisms drive the relationship in females than in males. In past work, with task-fMRI (Lai, Lombardo et al., 2019), we interpreted one of such mechanisms to be differing cognitive strategy that might be underpinned by an important vMPFC function – self-representation. The current work adds a new angle to what could possibly be the mechanism, as a second and non-mutually exclusive alternative explanation, could have something to do with cellular mechanisms affecting E:I balance in vMPFC. These different levels of explanations could line up together – that is, E:I imbalance in vMPFC causes disturbance in the development of self-representation, which impairs ability to engage in cognitive strategies that effectively permits camouflaging in males.

“More intact E:I balance in the vMPFC of autistic females may enable better vMPFC-related function (e.g., self-representation) and thus potentially better enable these individuals to camouflage social-communicative difficulties and cope in social situations.”

7) Subsection “Human fMRI data analysis” appears to state that the mean of the BOLD timeseries across voxel is taken first, from which H is computed. Doesn't it make more sense to compute H for each voxel and then average after? How different are these two situations? I don't have an intuition for how averaging timeseries affect H, but an analogous situation in spectral analysis is destructive interference, where two high-amplitude oscillations may combine to a flat signal due to a phase difference.

We elected to compute mean time-series within a region for several reasons that have to do with spatial smoothness, the spatial extent of effect for DREADD manipulations, and to harmonize the analysis procedures between mouse and human rsfMRI datasets. In addition to the spatial smoothness of the rsfMRI data itself, steps in the preprocessing pipeline introduce spatial smoothing to some extent. The preprocessing step of smoothing will also explicitly make spatially proximal voxels correlated. Thus, computing a mean time-series allows us to summarize the time-series of highly correlated spatially proximal voxels. Other considerations about the mouse rsfMRI data are also noteworthy here. The DREADD manipulation itself affects numerous spatially proximal voxels in the area underlying the injection site, rather than only affecting specific voxels. Therefore, the time-series data from the rsfMRI DREADD data in mice was extracted as a mean time-series from ROI voxels. To keep rsfMRI analyses as similar as possible across mouse and human rsfMRI datasets, we applied an identical approach of computing a mean time-series for the region. To further address the reviewer’s comment here, for the human data, if we compute H first in every voxel in region p32 and then average across voxels, the results remain statistically significant (*F(5,104)* = 7.54, *p* = 0.007), thus indicating that the primary inferences remain the same. We have now included this result in the revised manuscript.

“A similar sex-by-diagnosis interaction appeared when using another metric such as the intrinsic neural timescale (Watanbe, Rees and Masuda, 2019) (Figure 5—figure supplement 1) and when H was first calculated at each voxel and then averaged across voxels (Figure 5—figure supplement 2).”

8) Another method question: why not fit PLS directly with the 2-column matrix (CF1 and CF2) directly, since PLS is capable of outputting a single explanatory component, instead of performing PCA first. They should be similar?

Our application of PCA to extract the shared component of CF1 and CF2 was based on the idea of taking the same approach taken in prior work on the operationalization of camouflaging, where we use PCA on CF1 and CF2 to compute the CF measure (Lai et al., 2017; Lai, Lombardo et al., 2019). We have now added this detail to the revised manuscript.

“This method was utilized in order to be consistent with prior work which computed the camouflaging metric in an identical fashion (Lai et al., 2017; 2019).”

9) Similarly (and maybe I'm not understanding something here), why not fit PLS or mass univariate over the entire cortex for camouflaging as well, as the authors did to find spatial specificity for the sex*diagnosis contrast, instead of only regressing in vMPFC specifically?

We specifically assessed the camouflaging correlation in vMPFC specifically given that there was a sex*diagnosis interaction result and because of prior work that also found vMPFC-camouflaging correlations (Lai, Lombardo et al., 2019). This is indicated in the following sentences in the Results:

“In prior task-fMRI work we found a similar sex-by-diagnosis interaction in vMPFC self-representation response and a female-specific brain-behavioral correlation with camouflaging ability (Lai et al., 2019). Given that adult autistic females engage more in camouflaging on-average (Lai et al., 2017; Hull et al., 2019 and Schuck, Flores and Fung, et al., 2019), we next asked whether vMPFC H would be related to camouflaging in a sex-specific manner.”

Therefore, this was a hypothesis-driven follow-up analysis driven by observations in prior work finding camouflaging correlations in vMPFC and thus, did not warrant whole-brain testing or PLS.

10) Lastly, while I appreciate the ample reference to Gao et al., 2017, I suggest the authors check out Lombardi et al., 2017 as well, as they also look at how 1/f changes as a result of E:I shifts, but with a recurrent and interacting-EI spiking model that is arguably more similar to the model in this manuscript. However, their finding is in contrast to Gao et al. 2017, where 1/f flattens for increasing inhibition. This is worth referencing and noting in the discussions.

We thank the reviewer for pointing us to this paper. The paper is indeed of interest, as it shows with a simplified non-realistic neural model with discrete update dynamics that the 1/f slope of the PSD of avalanche activity is different when considering different percentages of inhibitory neurons. The limitations of Lombardi’s model in realism of neural dynamics do not allow us to relate their results directly to our findings. Also, the fact that they do not explicitly study variations in E:I parameters, but instead networks with different percentage of inhibitory neurons, does not provide us with the information we need for our study. We feel that for our work it would be more important to cite other papers (especially from Brunel, Wang and colleagues that profoundly influenced this field and our work) that show, using realistic models of neural dynamics, power spectra can be related directly to the frequency content of electrophysiological data. Interestingly, in those papers, Brunel and colleagues report effects that point in a different direction with respect to the results of Gao et al. This is one of the reasons that motivated us in the present study to reevaluate the results of the Gao et al. model with a recurrent network model.

To accommodate for the reviewer’s concerns, we improved introduction by adding the following references, including the Brunel and Wang paper as well as the Lombardi et al., paper, to the following sentence in Introduction:

“Models of neural networks have reported that the E:I ratio has profound effects on the spectral shape of electrophysiological activity (Brunel and Wang, 2033; Mazzoni et al., 2008 and Lombardi, Herrmann and de Arcangelis, 2017).”

In the Discussion, we added the following paragraph that we hope it will be useful for the reader, and motivate future work, regarding the interesting effects suggested by the reviewer.

“Results about the relationship between spectral shape and E:I balance obtained with our model of recurrent excitation and inhibition are largely compatible with those obtained with an earlier model of uncoupled excitation and inhibition (Gao, Peterson and Voytek, 2017). The uncoupled model predicts a linear increase of the slope value (i.e. flatter, less negative slopes) as E:I ratio increases. This is because in the uncoupled model changing the E:I ratio modifies only the ratio of the contribution to the LFP spectra of excitatory (faster time constant) and inhibitory (slower time constant) synaptic currents, leading to a linear relationship between slopes and E:I. In contrast, we found that the relationship between E:I and the spectral slope flattens out for high values of I. This, in our view, may in part arise from the fact that, as shown in studies of recurrent network models (Brunel and Wang, 2003), higher recurrent inhibition leads to higher peak frequency of γ oscillations (i.e. an increase of power at higher frequencies) thus partly counteracting the low-pass filtering effect of inhibitory currents in the uncoupled model. We plan to investigate in future studies how these opposing effects interact in a wider range of configurations and to use these results to gain a better understanding of the relationship between E:I ratio and LFP spectral shape.”

Reviewer #3:This is an impressive piece of work combining multi-level methods and different modalities using both mouse and human data that substantially enhances our understanding of E:I imbalance in autism. The authors demonstrate and extend the link between E:I balance and the Hurst component and 1/f slope in local field potentials and rsfMRI data using a biologically more plausible network model than prior work. They further establish the spatial distribution of autism-linked genes associated with excitation and androgen-sensitive genes that coincide with the sex-differential region associated with the H-component in the human rsfMRI data. Interestingly, this region also shows a sex-differential relationship to camouflaging behavior. This is an impressive set of analyses with elaborate and thorough methodological steps. Listed below are some minor comments that can help increase the clarity of the manuscript and interpretation of the results.1) The authors start the Introduction with the sentence that E:I imbalance affects many disorders. This calls for at least a short mention in the discussion how findings are specific to autism.

We have now inserted some text in the Discussion relevant to this point.

“This work may also be of broader relevance for investigating sex-specific E:I imbalance that affects other early-onset neurodevelopmental disorders with a similar male-bias as autism (Rutter, Caspi and Moffitt, 2003). For instance, conditions like ADHD affect males more frequently than females and also show some similarities in affecting behavioral regulation and associated neural correlates (Chantiluke et al., 2015). Furthermore, gene sets associated with excitatory and inhibitory neurotransmitters are linked to hyperactivity/impulsivity severity in ADHD, suggesting that E:I-relevant mechanisms may be perturbed (Naaijen, et al., 2017). It will be important for future work to test how specific sex-specific E:I imbalance is to autism versus other related sex-biased neurodevelopmental disorders.”

2) The Introduction lacks a bit of a red line to my taste. It would be clearer to read if sub-paragraphs were linked to each other. The different sections make sense – however it is left to the reader to link them to each other. Given the complexity and many aspects touched on, I think the flow of the Introduction could still be improved a bit.

We have now edited the writing of the Introduction to make the paragraphs more streamlined.

3) Could authors explain in more details what “in silico” refers to in this context? This might not be clear to a broad readership.

We have added “*(i.e. computational model)*” after the first instance of in silico in the manuscript.

“To achieve this aim, we first took a bottom-up approach by using in silico (i.e. computational) models of local neuronal microcircuitry to make predictions about how H and 1/f slope in local field potentials (LFP) and rsfMRI data may behave when there are underlying changes in E:I balance.”

4) How was mean frame-wise displacement calculated? Power? Jenkinson?

We have now added a reference to Power et al., in order to show how FD was calculated.

5) In Table 3, could authors also report differences in symptom scores and camouflaging between autistic males and females.

We have now inserted statistics for comparisons between autistic males and females on camouflaging and symptom scores in Table 3 (in the column labeled Sex).

6) What is the justification for the use of PLS on top? What additional information is revealed?

PLS was used as a multivariate alternative to contrast to mass-univariate testing. If effects are correlated across multiple brain regions, PLS will be able to identify such distributed systems-level effects that are multivariate in nature. The effects in the mass-univariate analysis that are subthreshold (potentially due to lower power for such subtle effects in a mass-univariate analysis framework) could be part of a more distributed neural system expressing the sex-by-diagnosis effect, especially given that many such regions are part of the DMN and other regions that show some level of overlap with the DHT-sensitive and ASD-excitatory gene map. Thus, PLS allows us to test and confirm this hypothesis.

“In contrast to mass univariate analysis, we also used partial least squares (PLS) analysis as a multivariate alternative to uncover distributed neural systems that express the sex-by-diagnosis interaction.”

7) A large part of the results is in mouse-data. The Introduction should thus also include a section introducing how E:I imbalance can be investigated in animal models. Right now, the Introduction refers solely to human participants, whereas the entire first Result section does not.

We have now included more text in the Introduction to discuss how techniques like chemogenetics in animals can be used to study E:I mechanisms, and how such animal work is key in translationally being able to bridge the gap between species.

“Next, our in-silico predictions are then tested in-vivo with a combination of rsfMRI and experimental manipulations in mice that either increase neurophysiological excitation or that silence the local activity in the network. Chemogenetic (i.e. designer receptors exclusively activated by designer drugs; DREADD) or optogenetic manipulations are optimally suited to these purposes, owing to the possibility of enabling remote control of neuronal excitability with cell-type and regional specificity (Yizhar et al., 2011; Ferenczi et al., 2016). Manipulations of neuronal activity like these in animals are key for two reasons. First, they allow for experimental in-vivo confirmation of in-silico predictions. Second, such work is a key translational link across species (i.e. rodents to humans), given the common use of neuroimaging readouts from rsfMRI (Balsters et al., 2020).”

8) Given the results-first format, it should be pointed out briefly which dataset was used for which set of results. The first section should clearly state that this is done in the mouse-data for example.

We have now tried to include more text to address this point.

“Changes in H in BOLD after chemogenetic manipulation to enhance excitability of excitatory neurons in mice

All of the results thus far report results from our in-silico model of recurrent neuronal networks and their readouts as simulated LFP or BOLD data. The in-silico modeling of BOLD data suggests that if E:I ratio is increased via enhanced excitability of excitatory neurons, then H should decrease. To empirically test this prediction in-vivo, we measured rsfMRI BOLD signal in prefrontal cortex (PFC) of mice under conditions where a chemogenetic manipulation (hM3Dq DREADD) (Alexander et al., 2009) is used to enhance excitability of pyramidal neurons.”

“Autism-associated genes in excitatory neuronal cell types in the human brain are enriched for genes that are differentially expressed by androgen hormones

The in-silico and in-vivo animal model findings thus far suggest that excitation affects metrics computed on neural time-series data such as 1/f slope and H. Applied to the idea of sex-related heterogeneity in E:I imbalance in autism, these results make the prediction that excitatory neuronal cell types would be the central cell type affecting such neuroimaging phenotypes in a sex-specific manner. To test this hypothesis about sex-specific effects on excitatory neuronal cell types, we examined whether known autism-associated genes that affect excitatory neuronal cell types (Satterstrom et al., 2020; Velmeshev et al., 2019) are highly overlapping with differentially expressed genes in human neuronal stem cells when treated with a potent androgen hormone, dihydrotestosterone (DHT)(Lombardo et al., 2018; Quartier et al., 2018).”

“H is on-average reduced in adult autistic men but not women

We next move to application of this work to human rsfMRI data in autistic men and women.”

9). The sex-differential results in humans are intriguing. However, when looking at Figure 5B, I wonder whether a gender-incoherent pattern is evident here? Could authors further discuss this?

We thank the reviewer for pointing out how the pattern looks like it might be similar to a gender-incoherence pattern. We have now amended the text to discuss how the effect could be compatible with an effect like gender incoherence. However, for stronger statements on this interpretation of the results, we would need to better understand how normative, population-level sex differences manifest in H and to better understand the normative ranges that H can take for each brain region, sex, and across age, socio-economic and socio-cultural factors (e.g., gender, race). This is something that future work with much larger normative, general population-based datasets could illuminate.

“The observed effect in vMPFC may also be consistent with a “gender-incoherence” pattern (i.e. towards reversal of typical sex differences in autism) (Bejerot et al., 2012). However, sex-specific normative ranges would need to be better established before interpreting effects in autism as being reversals of normative sex differences. More work with much larger general population-based datasets is needed to establish whether there are robust normative sex differences in H and to describe the normative ranges of H may take for each brain region, sex, and across age. Such work would also help with normative modelling (Bethlehem et al., 2018) approaches that would enable identification of which autistic individuals highly deviate from sex-specific norms.”

Also, in line with this, I wonder whether authors could refine this part of the Discussion stating: "Thus, one potential explanation for the male-specific reduction of H in vMPFC could have to do with early developmental and androgen-sensitive upregulation of genes that play central roles in excitatory neuron cell types, and thus ultimately affecting downstream E:I imbalance. Such effects may be sex-differential and thus less critical in human females, serving an important basis of sex-differential human brain development and explaining the sex-based heterogeneity and qualitative sex differences of autism neurobiology in human." Do authors refer to typical females here? The results look like TD females show a similar pattern to autistic males. So how do authors explain the differential pattern in TD females and autistic females.

We have now cleaned up the writing of this sentence to be more clear. Here we are simply referring to how androgen-dependent effects on gene expression within excitatory neurons in autism may be less critical in human females whom will not be exposed (on-average) to such high levels of androgens in early development.

“Such effects may be less critical in human females and may serve an important basis for sex-differential human brain development (Kaczkurkin, Raznahan and Satterthwaite, 2019). These effects may also help explain why qualitative sex differences emerge in autism (Lai et al., 2017; Bedford et al., 2019).”

Regarding the comparison of TD females to autistic males, we have refrained from making such cross-sex and cross-diagnostic comparisons because it is still not clear whether and how the range of normative variability in H can be defined in a sex-specific manner, in the absence of large-scale population-based dataset at this stage. The TD females vary more than TD males in region p32, and thus it may not be pertinent to compare autistic males to TD females given this difference in normative variability between sexes.

10) I like the link to the camouflaging behavior. However, the question arises whether there is also a sex-differential link to core autistic symptoms such as for example repetitive behaviors that have also been shown to differ across autistic males and females?

We have now analyzed data for ADOS and ADI-R RRB scales but could not identify any effects that would represent a sex-differential effect.

“Beyond this hypothesis-driven comparison of the relationship between H and camouflaging in vMPFC, we also ran correlations with ADI-R, ADOS and AQ scores. ADOS social-communication (SC) was negatively correlated with vMPFC H in autistic females (r = -0.51, p = 0.008) indicating higher H with lower SC severity. This relationship was not present in autistic males (r = -0.04, p = 0.83). However, the difference between these correlations was not statistically significant (z = 1.70, p = 0.08). ADI-R subdomains, ADOS RRB, and AQ correlations were not statistically significant.”

11) Could authors include information on the age range of autistic subjects (currently, I can't see the age range anywhere) and discuss the potential effect of age on their results from a neurodevelopmental perspective?

We have now inserted age ranges into the manuscript.

“Adult native English speakers (n=136, age range = 18-49 years) with normal/corrected-to-normal vision participated”

We have also inserted some text in the Discussion to talk about the effect of age.

“Similarly, future work should investigate how H may change over development. Prior work has shown that H and other related measures such as 1/f slope can change with normative and pathological aging in both rsfMRI and EEG data (Maxim et al., 2005; Wink et al., 2006 and Voytek et al 2015). Imperative to this work will be the establishment of age and sex-specific norms for H in much larger datasets. Age and sex-specific norms will enable more work to better uncover how these biomarkers may be affected in neurodevelopmental disorders or disorders relevant to neurodegeneration. Such work combined with normative modeling approaches (Bethlehem et al., 2018) may help uncover how experiential and environmental effects further affect such metrics.”

**References:**

Armbruster, B. N., Li, X., Pausch, M. H., Herlitze, S., and Roth, B. L. (2007). Evolving the lock to fit the key to create a family of G protein-coupled receptors potently activated by an inert ligand. *Proceedings of the National Academy of Sciences, 104(12)*, 5163-5168.

Atasoy, D., Betley, J. N., Su, H. H., and Sternson, S. M. (2012). Deconstruction of a neural circuit for hunger. *Nature, 488(7410)*, 172-177.

Carter, M. E., Soden, M. E., Zweifel, L. S., and Palmiter, R. D. (2013). Genetic identification of a neural circuit that suppresses appetite. *Nature, 503(7474)*, 111-114.

Coletta L, Pagani M, Whitesell JD, Harris JA, Bernhardt B, Gozzi A (2020) Network structure of the mouse brain connectome with voxel resolution. bioRxiv:2020.2003.2006.973164.

Ferguson, S. M., Eskenazi, D., Ishikawa, M., Wanat, M. J., Phillips, P. E., Dong, Y.,.… and Neumaier, J. F. (2011). Transient neuronal inhibition reveals opposing roles of indirect and direct pathways in sensitization. *Nature Neuroscience, 14(1)*, 22-24.

Krashes, M. J., Koda, S., Ye, C., Rogan, S. C., Adams, A. C., Cusher, D. S.,.… and Lowell, B. B. (2011). Rapid, reversible activation of AgRP neurons drives feeding behavior in mice. *The Journal of Clinical Investigation, 121(4)*, 1424-1428.

Liu X, Zhu XH, Zhang Y, Chen W (2011) Neural origin of spontaneous hemodynamic fluctuations in rats under burst-suppression anesthesia condition. Cereb Cortex 21:374-384.

Orth M, Bravo E, Barter L, Carstens E, Antognini JF (2006) The Differential Effects of Halothane and Isoflurane on Electroencephalographic Responses to Electrical Microstimulation of the Reticular Formation. Anesthesia and Analgesia 102:1709-1714.

Ray, R. S., Corcoran, A. E., Brust, R. D., Kim, J. C., Richerson, G. B., Nattie, E., and Dymecki, S. M. (2011). Impaired respiratory and body temperature control upon acute serotonergic neuron inhibition. *Science, 333(6042)*, 637-642.

Reimann HM, Niendorf T (2020) The (Un)Conscious Mouse as a Model for Human Brain Functions: Key Principles of Anesthesia and Their Impact on Translational Neuroimaging. Frontiers in Systems Neuroscience 14.

Roth, B. L. (2016). DREADDs for Neuroscientists. *Neuron, 89,* 683-694.

Sasaki, K., Suzuki, M., Mieda, M., Tsujino, N., Roth, B., and Sakurai, T. (2011). Pharmacogenetic modulation of orexin neurons alters sleep/wakefulness states in mice. *PLoS One, 6(5)*, e20360.

Whitesell JD et al. (2020) Regional, layer, and cell-class specific connectivity of the mouse default mode network. bioRxiv:2020.2005.2013.094458.